# Krill and salp faecal pellets contribute equally to the carbon flux at the Antarctic Peninsula

Nora-Charlotte Pauli [1,2✉], Clara M. Flintrop [2,3], Christian Konrad[2,3], Evgeny A. Pakhomov[4,5,6], Steffen Swoboda [3], Florian Koch[2], Xin-Liang Wang[7], Ji-Chang Zhang[7], Andrew S. Brierley [8], Matteo Bernasconi[8], Bettina Meyer [1,2,9✉] & Morten H. Iversen [2,3✉]

Krill and salps are important for carbon flux in the Southern Ocean, but the extent of their contribution and the consequences of shifts in dominance from krill to salps remain unclear. We present a direct comparison of the contribution of krill and salp faecal pellets (FP) to vertical carbon flux at the Antarctic Peninsula using a combination of sediment traps, FP production, carbon content, microbial degradation, and krill and salp abundances. Salps produce 4-fold more FP carbon than krill, but the FP from both species contribute equally to the carbon flux at 300 m, accounting for 75% of total carbon. Krill FP are exported to 72% to 300 m, while 80% of salp FP are retained in the mixed layer due to fragmentation. Thus, declining krill abundances could lead to decreased carbon flux, indicating that the Antarctic Peninsula could become a less efficient carbon sink for anthropogenic $CO_2$ in future.

[1] Institute for Chemistry and Biology of the Marine Environment (ICBM), Carl-von-Ossietzky University of Oldenburg, Carl-von-Ossietzky-Str. 9-11, 26111 Oldenburg, Germany. [2] Alfred Wegener Institute, Helmholtz Centre for Polar and Marine Research, Am Handelshafen 12, 27570 Bremerhaven, Germany. [3] MARUM and University of Bremen, Leobener Str. 8, 28359 Bremen, Germany. [4] Department of Earth, Ocean, and Atmospheric Sciences, University of British Columbia, 2207 Main Mall, Vancouver, British Columbia V6T 1Z4, Canada. [5] Institute for the Oceans and Fisheries, University of British Columbia, 2202 Main Mall, Vancouver, British Columbia V6T 1Z4, Canada. [6] Hakai Institute, PO Box 25039 Campbell River, British Columbia V9W 0B7, Canada. [7] Yellow Sea Fisheries Research Institute, Chinese Academy of Fishery Sciences, 106 Nanjing Road, Qingdao 266071, China. [8] Pelagic Ecology Research Group, Gatty Marine Laboratory, Scottish Oceans Institute, School of Biology, University of St Andrews, St Andrews, Fife KY16 8LB, UK. [9] Helmholtz Institute for Functional Marine Biodiversity (HIFMB), Ammerländer Heerstraße 231, 26129 Oldenburg, Germany. ✉email: nora-charlotte.pauli@awi.de; bettina.meyer@awi.de; morten.iversen@awi.de

The Southern Ocean (SO) is a significant carbon sink that accounts for about 40% of the global ocean uptake of anthropogenic $CO_2$ and ~50% of the total atmospheric uptake[1,2]. The bulk of the primary production is recycled in the epipelagic zone[3]. In high latitudes, ~15–25% is exported below 200 m[4] and only ~10% of this export flux eventually reaches depths below 1000 m, where carbon is sequestered for >100 years[3]. This flux of particulate organic carbon (POC) from the surface to the deep ocean is mediated by the biological carbon pump (BCP)[5]. The efficiency of the BCP is driven by sinking organic aggregates, while fragmentation and grazing by zooplankton and microbial remineralisation processes decrease the efficiency of the BCP[6]. One major constituent of sinking aggregates are zooplankton faecal pellets (FP), which sink at high velocities and can make up the vast majority of sinking particles locally and therefore play a crucial role in carbon export[7,8]. In addition, vertical migrations of zooplankton and the subsequent production of FP below the mixed layer contributes to the export of carbon[9].

Antarctic krill, *Euphausia superba* Dana, 1850 (hereafter krill), is a key species in the SO, both in terms of representing the main trophic link between primary producers and apex predators[10], as well as for biogeochemical cycles via a high production of large, carbon-rich and fast sinking FP[11,12]. The contribution of krill FP to the carbon flux varies and ranges up to 281 mg C m$^{-2}$ d$^{-1}$[13]. At the Western Antarctic Peninsula (WAP) and the marginal ice zone, krill FP were shown to account for 17–72% of the total carbon flux[14,15]. The contribution of krill to the carbon flux depends on their abundance, the carbon content and sinking velocity of their FP, as well as on fragmentation and degradation processes of FP during their descent through the water column[11,14,16]. This contribution can be disproportionally high when krill form dense, extensive swarms, which produce a rain of FP, possibly exceeding the grazing efficiency of the available detritivores[14,16]. This is particularly evident at the WAP, where krill occur at high abundances[17], and their FP can account for >80% of the sinking particles[18] and for >60% of the organic carbon in sea floor sediments[19].

Over the past decades, the WAP region has experienced dramatic climatic changes, including a 6 °C increase in mean winter air temperature since 1950, causing a 10% decline in sea ice extent[20,21], making the WAP one of the most rapidly warming regions worldwide[22]. In response to these climatic changes, krill abundances have declined north of 60°S and shifted southward since 1926[23]. At the same time, the warming temperatures have allowed the pelagic tunicate *Salpa thompsoni* Foxton, 1961 (hereafter salp) to expand their distribution with increasing abundances in the SO, particularly around the WAP[24], extending into previously krill dominated areas[25]. Salps produce large FP, which settle at 2.5-fold higher average velocities than krill pellets (760 vs. ~300 m d$^{-1}$)[11,26]. Salp FP contribute between 5 and 66% to the total flux of POC[15] and were shown to increase the local carbon export up to 10-fold compared to non-salp areas[27]. In addition, salps can feed at high efficiencies on a larger size range of particles than krill[28,29], and are able to respond rapidly to favourable conditions by asexual production and the formation of dense swarms[30]. This may allow salps to package more plankton into FP than krill[8], with a shift from krill to salps thus resulting in a more efficient carbon export in the SO.

Recent studies have investigated the separate contributions of krill and salp FP to the carbon flux[14,31], while studies directly comparing both species are lacking. Yet, this comparison is crucial to assess the implications of a possible long-term shift from krill to salps as dominant grazers at the WAP[24,32]. Krill and salps differ largely in their feeding mechanism, life cycle and in their ecological niches, thus a long-term shift might have cascading effects on the SO food web and biogeochemical cycles. Despite the high sinking velocities of salp FP, it was recently shown that only 13% of the salp pellets produced in the upper 100 m reached sediment traps at 300 m[31]. It was hypothesised that zooplankton feeding on salp FP might cause them to break into smaller, slow sinking particles, potentially increasing their recycling in the surface ocean[31]. Thus, intact krill FP are more frequently found in sediment traps[33], suggesting that they are transferred to deeper water layers at high efficiency[18] and thus are more efficient in carbon export than salp FP. To assess the consequences of a shift from krill to salps it is therefore pivotal to shed more light on the comparative contribution of krill and salp FP to the carbon cycle in the SO.

Previous studies have used modelling approaches, sediment traps or moorings, and/or measurements of FP production and FP sinking velocities to estimate the FP carbon flux[14,31,34]. In addition, the use of viscous gels in sediment traps (hereafter gel traps), which preserve the shape, size and structure of sinking particles, in combination with traditional sediment traps and in-situ particle camera systems, have provided high resolution vertical profiles of particles sizes and abundance[6,35]. The use of gel traps also preserve salp FP[31], which are fragile and easily break apart in conventional sediment traps, preventing direct flux estimates for salp FP. However, a combination of these in-situ approaches with measurements of biomass, FP production, sinking velocities, and microbial degradation is needed to provide better estimates of the extent of FP carbon flux.

Here, we provide a direct comparison of the contribution of krill and salp FP to the total organic carbon flux along Elephant Island at the northern tip of the Antarctic Peninsula. We do this by combining a series of five consecutive drifting sediment traps equipped with conventional and gel traps with vertical in-situ particle camera profiles at high temporal resolution. This is accompanied by on-board measurements of FP production rates, FP size-specific sinking velocities, and microbial degradation. Moreover, we use zooplankton net tows and hydroacoustics to determine in-situ abundances of krill and salps. Combining measurements of primary production, standing stock of POC and carbon content of FP produced by both species allows for insights into the comparative role of krill and salps in the biological carbon pump. We show that krill and salp FP contribute equally to the carbon flux at 300 m. In contrast to previous assumptions, krill FP are exported more efficiently to 300 m than salp FP, which are mostly retained in the top 200 m. Thus, a long-term, shift in dominance from krill to salps could lead to a less efficient carbon sink in the future SO.

## Results

**Standing stock of chlorophyll *a* and particulate organic carbon.** Average chlorophyll *a* concentration in the study area was below 1 mg m$^{-3}$ and the majority of the standing stock of chlorophyll was found between the surface and 200 m (Table 1; Supplementary Fig. 1). The average integrated standing stock of chlorophyll ranged from 64.8 to 115.8 mg m$^{-2}$ and from 102.7 to 184.9 mg m$^{-2}$ for the 0–100 m and 0–200 m depth layers, respectively (Table 1). Similarly, the standing stock of POC was concentrated in the top 200 m with an integrated average of 12.45 ± 4.2 g m$^{-2}$. Dinoflagellates and diatoms dominated the ambient plankton community[36], at a mean temperature of 0.6 °C and a mean salinity of 34.4 in the upper 200 m. Primary production at 20 m in the study area at Elephant Island was 20.62 mg C m$^{-3}$ d$^{-1}$ (103.1 mg C m$^{-2}$ d$^{-1}$). In comparison, the mean primary production across different areas around the northern Antarctic Peninsula in the same month (04/2018) was 10.4 ± 9.5 mg C m$^{-3}$ d$^{-1}$.

**Table 1 Standing stock of chlorophyll *a* (Chl. a), primary production (PP), particulate organic carbon (POC), particulate organic nitrogen (PON), and the carbon to nitrogen ratio (C:N) integrated over the top 200 m.**

|  | DF 1 | DF 2 | DF 3 | DF 4 | DF 5 | Mean ± SD |
|---|---|---|---|---|---|---|
| Chl. *a* [mg m$^{-2}$] | 102.7 | 109.5 | 127.9 | 184.9 | 131.3 | 131.3 ± 37.4 |
| PP [mg C m$^{-2}$ d$^{-1}$] | – | – | 103.1 | – | – | 103.1 ± 0 |
| POC [g m$^{-2}$] | 9.8 | 9.2 | 18.3 | 12.5 | 12.5 | 12.5 ± 4.2 |
| PON [g m$^{-2}$] | 1.3 | 1.2 | 1.6 | 2.0 | 1.5 | 1.5 ± 0.4 |
| C:N | 9.1 | 8.8 | 13.4 | 7.4 | 9.7 | 9.7 ± 2.6 |

Values are given for each of the respective stations and drifting trap deployments (DF 1–5). The mean and standard deviation (SD) is given across all stations for each parameter, respectively.

**Table 2 Abundances of krill and salps at day and night.**

| Abundance [Ind. m$^{-2}$] |  | DF 1 | DF 2 | DF 3 | DF 4 | DF 5 | Mean | SD |
|---|---|---|---|---|---|---|---|---|
| Krill | Day | 29.20 | 81.60 | 51.50 | 60.20 | 3117.00 | 667.90 | ± 1369.22 |
|  | Night | 505.00 | 175.00 | 76.90 | 135.00 | 56.40 | 189.66 | ± 182.42 |
| Salps | Day |  |  |  |  |  | 93.9 | ± 149.3 |
|  | Night |  |  |  |  |  | 278.9 | ± 87.7 |

Abundances are shown for each of the five drifting trap deployments (DF 1–5), as well as a mean across all traps with standard deviation (SD). Salp (*Salpa thompsoni*) abundances are based on quantitative oblique net tows and integrated over 170 m. Krill (*Euphausia superba*) abundances are based on the hydroacoustic survey integrated over the top 200 m. Day and night were defined according to the local time of sunrise and sunset as the periods from 06:00 am to 19:00 pm, and from 19:00 p.m. to 06:00 a.m., respectively (UTC –03:00).

**Table 3 Faecal pellet (FP) parameters for Antarctic krill (*Euphausia superba*) and salps (*Salpa thompsoni*).**

|  | Krill |  |  | Salps |  |
|---|---|---|---|---|---|
| FP Production [mm³ h$^{-1}$ Ind.$^{-1}$] | 0.06 | ± 0.05 |  | 0.629 | ± 0.732 |
| FP Production [mg C h$^{-1}$ Ind.$^{-1}$] | 0.004 | ± 0.004 |  | 0.016 | ± 0.019 |
| FP carbon/Volume [mg C mm$^{-3}$] | 0.080 | ± 0.044 | Type 1 | 0.026 | ± 0.012 |
|  |  |  | Type 2 | 0.017 | ± 0.008 |
| FP sinking velocity [m d$^{-1}$] | 233.4 | ± 154.3 |  | 586.0 | ± 692.0 |

Salp abundances are based on quantitative oblique net tows and integrated over 170 m. Krill abundances are based on the hydroacoustics survey integrated over the top 200 m. For FP, the volume and carbon–specific production rates and sinking velocities are given for krill and salps, respectively. FP carbon content is shown for krill FP and the two types of salp FP (type 1 = phytoplankton, type 2 = ingested krill FP). Values are given as mean across all sediment trap deployments. The SD is shown for each value.

**Abundances of krill and salps**. Salp abundances were obtained from quantitative net hauls using Isaacs–Kidd Midwater Trawls (IKMT) and Rectangular Midwater Trawls (RMT) in the upper 170 m of the water column. Abundances were higher at night (278.98 ± 87.7 Ind. m$^{-2}$; Table 2) than during daytime (93.96 ± 149.3 Ind. m$^{-2}$). Aggregate salps (sexual generation, blastozooids) with a size of 7–20 mm (oral–atrial body length = OAL) dominated the observed salp population. Solitary salps (asexual generation, oozooid) showed a lower abundance and were mainly represented by individuals of > 50 mm OAL. The biomass of krill (g m$^{-2}$) was obtained from the hydroacoustic survey and converted to abundance (Ind. m$^{-2}$) using a representative length–weight relationship[37]. Krill abundance integrated over the top 200 m was 189.66 ± 182.4 Ind. m$^{-2}$ at night and 667.9 ± 1396.2 Ind. m$^{-2}$ during daytime (Table 2). Single events of large krill aggregations reached abundances of 4,076 to 195,232 Ind. m$^{-2}$. The average size of krill (total body length, AT), obtained from quantitative net hauls, was 42.9 ± 6.4 mm, with a dominance of female krill (63.3%) and few juveniles (3.7%).

**Faecal pellet production, carbon content and sinking velocities**. Individual FP production rates of krill measured from incubation experiments were 0.06 ± 0.05 mm³ Ind.$^{-1}$ h$^{-1}$, corresponding to a carbon FP production (FPprod) of 0.004 ± 0.004 mg C Ind.$^{-1}$ h$^{-1}$ (Table 3; Supplementary Fig. 2). FP production rates as pellet carbon (Krill FP$_C$, [mg C Ind.$^{-1}$ h$^{-1}$]) increased with krill size (Length$_{Krill}$; Eq. 1).

$$\text{Krill FP}_C = 2E - 08\ \text{Length}_{Krill}^{3.2859},\ R^2 = 0.28 \tag{1}$$

The individual FPprod rates of salps were one order of magnitude higher than those of krill (0.63 ± 0.73 mm³ Ind.$^{-1}$ h$^{-1}$), corresponding to a carbon FPprod of 0.016 ± 0.019 mg C Ind.$^{-1}$ h$^{-1}$ (Table 3; Supplementary Fig. 2). We used the mean FPprod after 4–8 h of incubation, which is considered to most adequately represent in-situ production rates[38]. Similar to krill, salp FPprod rates (Salp FP$_C$, [mg C Ind.$^{-1}$ h$^{-1}$]) increased with increasing salp size (OAL$_{Salp}$; Eq. 2).

$$\text{Salp FP}_C = 0.0004\ \text{OAL}_{Salp}^{1.3309},\ R^2 = 0.63 \tag{2}$$

Krill FP had an average volume of 0.47 ± 0.48 mm³ and were smaller compared to salp FP (1.56 ± 1.73 mm³), while salp FP volume increased with salp size. Krill FP had a higher carbon to volume ratio than salp pellets (0.08 vs. 0.03 mg C mm$^{-3}$; Table 3). The POC to volume ratio for krill FP was slightly higher than what was previously reported (0.02–0.06 mg C mm$^{-3}$; see Supplementary Information)[15,16]. Based on krill and salp abundances and FPprod rates, salps packed 0.33–0.65% of the standing stock of POC (<200 m) into their FP per day, compared to 0.04–1.53% for krill. In addition, salp FP showed an average sinking velocity of 586.0 ± 692 m d$^{-1}$, which was about 3-fold higher than the sinking velocity of krill FP (233.38 ± 154.33 m d$^{-1}$; Table 3; Supplementary Fig. 3). Other sinking particles (marine snow) measured during this study only had an average sinking velocity

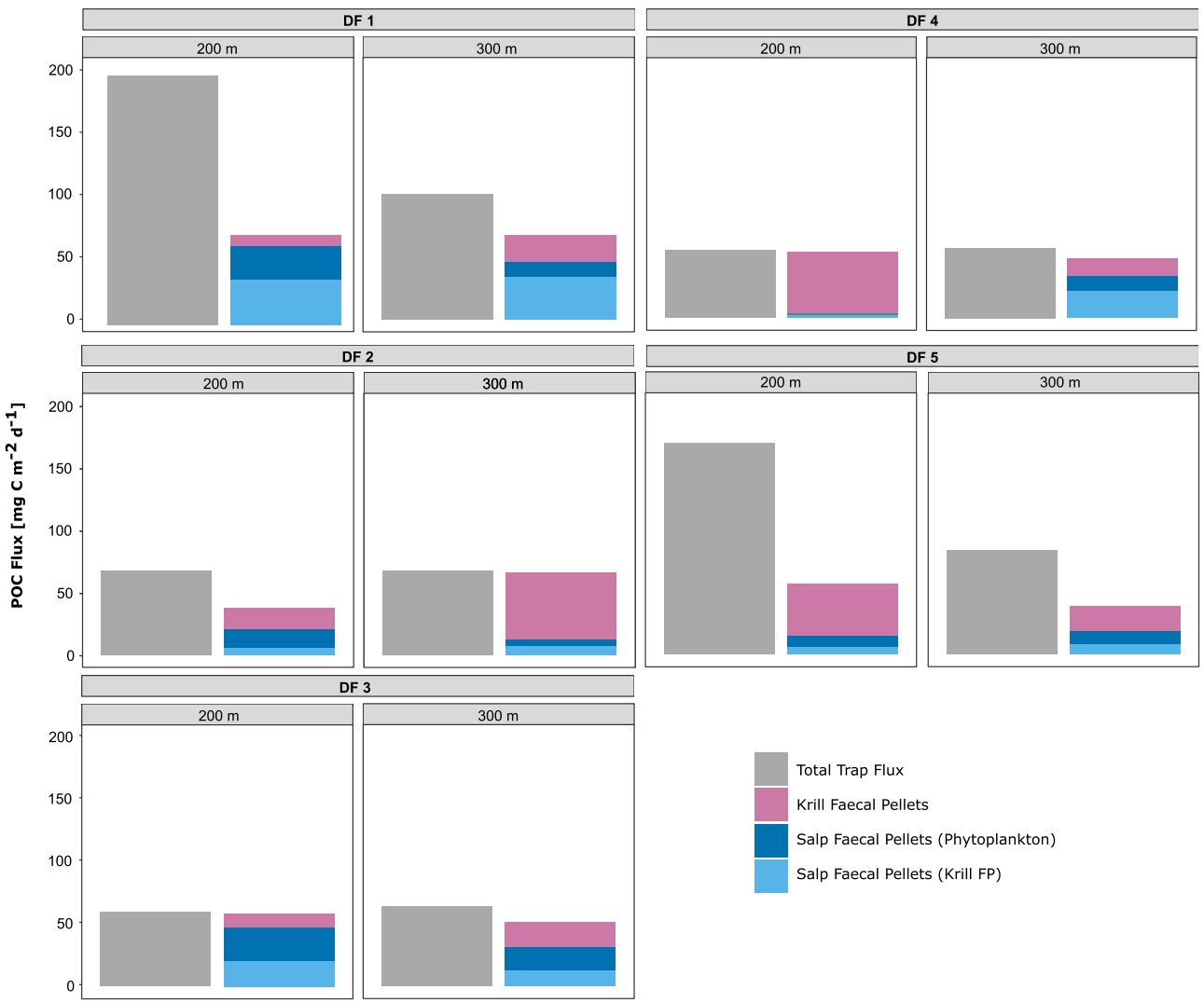

**Fig. 1 Flux of particulate organic carbon (POC) in the drifting traps.** The flux of POC is shown in mg C m$^{-2}$ d$^{-1}$. The panels depict the POC flux for the five drifting traps (DF 1 to 5) deployed at 200 and 300 m, respectively for 24 h each. The total POC flux is depicted in grey, and the respective flux of faecal pellets (FP) is shown in pink for krill FP, in dark blue for salp FP consisting of phytoplankton, and in light blue for salp FP consisting of ingested krill FP.

of $139 \pm 168$ m d$^{-1}$, highlighting the exceptionally high sinking velocity of krill and salp FP.

**Microbial respiration of pellet-associated microbes.** The oxygen gradient through the pellet-water interface was measured to determine the respiration rate of microbes associated with the pellets. The oxygen consumption was converted to carbon dioxide ($CO_2$) production by using a respiratory quotient of 1 mol $O_2$ to 1 mol $CO_2$. The carbon respiration was normalised to the total POC content of the individual FP to calculate the carbon-specific respiration (degradation) rates, which were $0.3 \pm 0.2\%$ d$^{-1}$ for salp FP ($n = 18$), and $1.2 \pm 0.6\%$ d$^{-1}$ for krill FP ($n = 4$).

**Flux of POC and faecal pellets measured with drifting traps.** We deployed five drifting traps consecutively over five days for ~24 h each (Supplementary Table 1). The total average flux of POC was $110.2 \pm 68.9$ mg C m$^{-2}$ d$^{-1}$ at 200 m and $74.5 \pm 17.6$ mg C m$^{-2}$ d$^{-1}$ at 300 m (Fig. 1; Table 4), with significant differences between the single trap deployments ($p = 0.04$, Kruskal−Wallis test, Df = 4, Chi−square statistic = 10). Krill and salp FP were found in all traps and at all depths (Supplementary Fig. 4). The combined contribution of krill and

salp FP to the total POC flux ranged from 33.5 to 98% and increased with increasing depth, accounting for $59.1 \pm 18.6\%$ at 100 m and $75.2 \pm 19.4\%$ at 300 m. The total flux of salp FP ranged from 3 to 63.7 mg C m$^{-2}$ d$^{-1}$ at 200 m and from 12.5 to 45.9 mg C m$^{-2}$ d$^{-1}$ at 300 m (Table 4). The flux of krill FP ranged from 8.5 to 50.2 mg C m$^{-2}$ d$^{-1}$ at 200 m and from 13.8 to 53.2 mg C m$^{-2}$ d$^{-1}$ at 300 m.

We observed two different types of salp FP in the sediment traps: Pellets composed of phytoplankton with a greenish colour (type 1), and darker pellets, composed of ingested krill pellets (type 2). Consistent with this, we frequently observed ingested krill FP in the stomachs of freshly caught salps (Supplementary Fig. 5). Type 1 FP had a higher carbon content than type 2 FP ($0.026 \pm 0.012$ vs. $0.017 \pm 0.008$ mg C mm$^{-3}$). The trap flux of type 2 pellets was slightly higher at 300 m compared to type 1 pellets (16.49 vs. 11.92 mg C m$^{-2}$ d$^{-1}$; Table 4), at 200 m the contribution of both salp FP types was nearly equal (type 1: 15.5 mg C m$^{-2}$ d$^{-1}$, type 2: 14.1 mg C m$^{-2}$ d$^{-1}$).

**Potential faecal pellet flux of krill and salps.** To determine the recycling, i.e. flux attenuation, of krill and salp FP, we calculated the potential FP flux based on the abundances of krill and salps

**Table 4 Carbon flux measured from the drifting sediment traps.**

| Flux [mg C m$^{-2}$ d$^{-1}$] | FP Type | Depth [m] | DF 1 | DF 2 | DF 3 | DF 4 | DF 5 | Mean | SD |
|---|---|---|---|---|---|---|---|---|---|
| Trap flux | | | | | | | | | |
| Total POC flux | | 100 | 90.04 | 98.85 | 47.26 | 50.44 | 58.78 | 69.07 | ±23.74 |
| | | 200 | 199.93 | 67.55 | 59.79 | 54.42 | 169.27 | 110.19 | ±68.95 |
| | | 300 | 100.68 | 67.77 | 64.00 | 56.62 | 83.50 | 74.51 | ±17.62 |
| Salp FP flux | 1 | 100 | 2.50 | 17.99 | 14.01 | 1.00 | 2.58 | 7.62 | ±7.81 |
| | 2 | | 0 | 11.02 | 2.85 | 2.94 | 2.61 | 3.88 | ±4.17 |
| | 1 + 2 | | 2.50 | 29.01 | 16.86 | 3.94 | 5.19 | 11.50 | ±11.33 |
| | 1 | 200 | 27.32 | 14.64 | 26.56 | 0.78 | 8.41 | 15.54 | ±11.51 |
| | 2 | | 36.34 | 5.69 | 20.32 | 2.24 | 5.94 | 14.10 | ±14.24 |
| | 1 + 2 | | 63.66 | 20.33 | 46.88 | 3.02 | 14.34 | 29.65 | ±24.92 |
| | 1 | 300 | 12.32 | 5.65 | 19.02 | 12.45 | 10.16 | 11.92 | ±4.82 |
| | 2 | | 33.67 | 6.82 | 11.99 | 21.75 | 8.20 | 16.49 | ±11.24 |
| | 1 + 2 | | 45.99 | 12.48 | 31.01 | 34.19 | 18.35 | 28.40 | ±13.27 |
| Krill FP flux | | 100 | 35.05 | 16.19 | 25.26 | 26.76 | 28.85 | 26.42 | ±6.83 |
| | | 200 | 8.47 | 17.70 | 11.68 | 50.23 | 42.36 | 26.09 | ±18.95 |
| | | 300 | 21.69 | 53.24 | 20.62 | 13.80 | 20.16 | 25.90 | ±15.59 |

The carbon flux in mg C m$^{-2}$ d$^{-1}$ was measured for each of the five drifting sediment trap deployments (DF 1–5) across the three deployment depths 100, 200 and 300 m is given as total particulate organic carbon (POC) flux. The respective contribution of krill faecal pellets (FP), and the two salp FP types (type 1 consisting of phytoplankton, type 2 consisting of ingested krill FP) to the carbon flux is shown, respectively. In addition to the values for each DF, a mean across all traps is shown with standard deviation (SD).

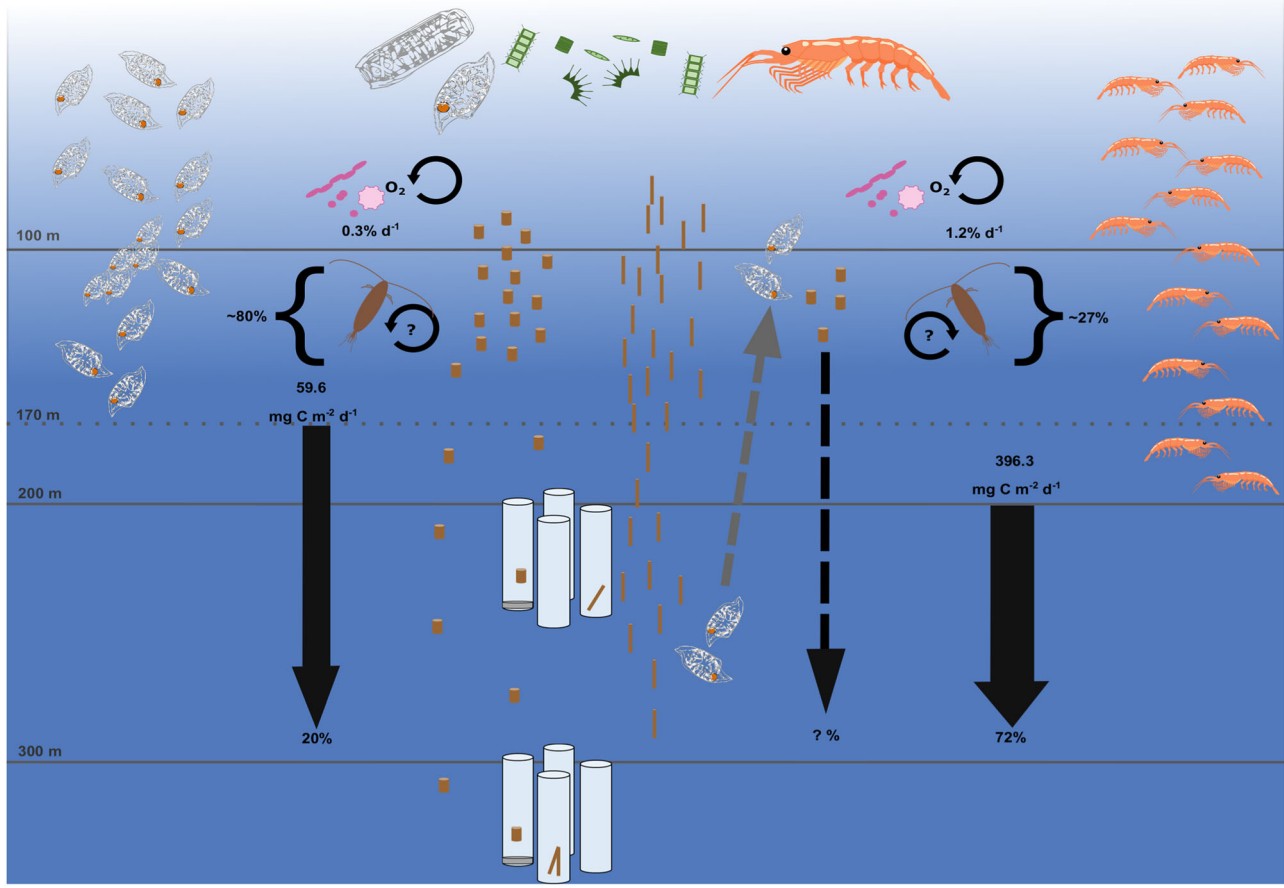

**Fig. 2 Potential faecal pellet (FP) flux and export efficiency of krill and salp FP.** The potential flux of FP of krill in the upper 200 m accounted on average for 396.3 mg C m$^{-2}$ d$^{-1}$, incl. the presence of dense krill swarms, resulting in an export efficiency of krill FP of 72% to 300 m. The flux of salp FP in the upper 170 m accounted for 59.6 mg C m$^{-2}$ d$^{-1}$ for FP produced from feeding on phytoplankton (type 1), resulting in an export efficiency of 20% to 300 m. The carbon-specific microbial respiration accounted for only 0.3% of the carbon loss per day for salp FP, and for 1.2% d$^{-1}$ for krill FP. Consequently, other remineralisation processes and zooplankton-mediated processes, such as feeding and loosening of the pellets must account for the retention of about 80% of salp FP, and 27% of krill FP in the top 200 m. The dashed arrow depicts the production of type 2 FP at the surface after feeding on krill FP at depth and the unknown contribution to the export flux of carbon.

**Table 5 Potential faecal pellet (FP) flux for krill and salps.**

| Species | Day/Night | Depth [m] | DF 1 | DF 2 | DF 3 | DF 4 | DF 5 | Mean | SD |
|---|---|---|---|---|---|---|---|---|---|
| Salps | Day | >170 | | | | | | 19.84 | ±29.82 |
| Salps | Night | >170 | | | | | | 39.78 | ±11.52 |
| Salps | 24 h | | | | | | | 59.62 | ±14.09 |
| Krill | Day | >200 | 3.14 | 24.29 | 3.10 | 3.62 | 1892.55 | 385.34 | ±842.61 |
| Krill | Night | >200 | 32.71 | 8.58 | 3.91 | 6.87 | 2.76 | 10.97 | ±12.37 |
| Krill | 24 h | | 35.85 | 32.87 | 7.01 | 10.49 | 1895.31 | 396.30 | ±838.07 |

The potential faecal pellet (FP) flux in mg C m$^{-2}$ d$^{-1}$, based on the in−situ abundances of krill and salps, is shown as mean across all drifting trap deployments (DF 1–5) for salps and for each individual trap deployment for krill. The mean potential FP flux across all drifting traps is shown along with the standard deviation. Salp abundances are based on quantitative oblique net tows and integrated over 170 m. Krill abundances are based on the hydroacoustics survey integrated over the top 200 m.

and compared it to the direct measurements of the FP flux obtained from the drifting traps. For salps, we multiplied the integrated abundance for each salp size with the size-specific FP carbon production rates for type 1 pellets (mg C Ind.$^{-1}$ h$^{-1}$, Eq. 2) to estimate the total daily carbon FPprod for each salp size-class in the upper 170 m. The individual carbon FPprod per salp size-class was subsequently added up to obtain the carbon FPprod of the whole population for day and night, respectively. The potential daily flux of type 1 salp FP (phytoplankton) was 59.62 ± 14.09 mg C m$^{-2}$ d$^{-1}$ (Fig. 2, Table 5), with a 2-fold higher pellet production at night compared to daytime (39.78 ± 11.52 vs. 19.84 ± 29.82 mg C m$^{-2}$ d$^{-1}$). This potential FP flux was compared to the trap flux of type 1 salp FP, which resulted in an average export efficiency of type 1 FP of 26.1% to 200 m, and 20% to 300 m. The export efficiency to 300 m varied between the individual trap deployments between 9.5% and 32%. Type 2 salp FP were not included in the export efficiency calculation as we assume that they result from salps feeding on krill FP during daytime at greater depth and are excreted at night when salps migrate to the surface to feed on phytoplankton, based on a gut passage time of about 8 h[38]. This pattern might decrease the efficiency of carbon export by krill FP. However, currently we cannot provide an accurate estimate of the impact of this coprophagous feeding behaviour on the efficiency of the biological carbon pump.

For krill, we used the mean size of krill obtained from the net tows in the study area and the regression of carbon FPprod (mg C Ind.$^{-1}$ h$^{-1}$; Eq. 1) multiplied by the krill abundance integrated over the top 200 m for day, night, and each drift trap deployment respectively. The flux of krill FP showed large differences between the single trap deployments, as well as between the three deployment depths (100, 200 and 300 m; Table 4), which was caused by the patchy distribution of krill and the occurrence of few vast swarms at three of the five drift trap deployments (DF 1, 2 and 5). The potential FP flux of krill in the upper 200 m during the deployments of drifting traps 1–4 showed no significant difference between night (13.02 ± 13.27 mg C m$^{-2}$ d$^{-1}$) and daytime (8.54 ± 10.51 mg C m$^{-2}$ d$^{-1}$). During the deployment of DF 5, the presence of a vast krill swarm resulted in a potential FP flux of 1892.6 mg C m$^{-2}$ d$^{-1}$ during day and 2.8 mg C m$^{-2}$ d$^{-1}$ (Table 5) at night. This resulted in an average export efficiency of 72.3% to 300 m. On average, the potential FP flux of krill and salps accounted for 48.3% and 57.8% of the primary production, respectively. For the calculation of the average export efficiency for both species, efficiencies of >100% were put to 100%.

**Krill pellet attenuation measured from the particle camera.** During the five drifting trap deployments, we deployed an in-situ camera system near the trap position to quantify the vertical abundance and size-distribution of sinking particles in ~4-h intervals. This resulted in a total of 38 profiles with a vertical resolution of 15 cm (Supplementary Table 4). We identified the volume concentration of krill FP from the camera profiles, which supported the patchy vertical and temporal distribution of krill FP that was observed from both the sediment traps and the calculated potential FP flux (Fig. 3). The camera profiles revealed dense clouds of krill FP between 300 and 500 m depth during the first trap deployment (DF 1), which correlated to a high krill abundance in the top 200 m and a high potential krill FP flux (505 Ind. m$^{-2}$, Table 2; 35.8 mg C m$^{-2}$ d$^{-1}$, Table 5). Similarly, dense clouds were observed between the surface and 250 m during the last trap deployment (DF 5), corresponding to the occurrence of a dense krill swarm during daytime and a high potential flux of krill FP (3117 Ind. m$^{-2}$; Table 2; 1892.6 mg C m$^{-2}$ d$^{-1}$; Table 5). Low in−situ concentrations of krill FP were observed during the deployments of drifting traps 3 and 4, where the krill standing stock at 0–200 m depth and thus the potential FP flux were low (51 to 135 Ind. m$^{-2}$; Table 2; 7 to 10.5 mg C m$^{-2}$ d$^{-1}$; Table 5).

## Discussion

In this study, we assess the contribution of krill and salp FP to the carbon flux at the Antarctic Peninsula (AP) and present a direct comparison. We found that together, krill and salp FP accounted for the majority of the total POC flux. Krill FP are efficiently exported out of the euphotic zone, while salp FP are subject to high turnover in the upper 200 m.

The total flux of POC at Elephant Island was 110 mg C m$^{-2}$ d$^{-1}$ at 200 m, coinciding with previous studies from the AP, reporting flux between 0 and 463 mg C m$^{-2}$ d$^{-1}$[18,34]. Zooplankton FP have been shown to play a significant role in the biological carbon pump[7,8] and at times account for >65% of the total POC flux at the AP[15,18]. In our study, krill and salp FP contributed similarly (64%) to the total POC flux at 200 m, increasing to 75% at 300 m. A relatively deep vertical mixing layer and high standing stock of chlorophyll a down to 200 m (Supplementary Fig. 1) and high abundances of krill and salps below 100 m (Supplementary Figs. 6, 7) indicated that feeding on phytoplankton and FP production was not limited to the upper 100 m.

Salps packed more of the daily primary production into their FP and converted 4-fold more carbon into their FP per hour than krill (16.3 vs. 4.5 µg C Ind.$^{-1}$ h$^{-1}$). In addition, the average sinking velocities of salp FP were 2.5-fold higher compared to krill FP. Consequently, we would expect salp FP to contribute more to the total POC flux than krill FP and to have high export efficiencies, as suggested in previous studies[39,40]. However, we found that on average krill and salp FP contributed equally to the total POC flux at 200 and 300 m. Hence, krill FP were exported at markedly higher efficiencies of ~72% to 300 m, while only ~20% of the salp FP consisting of phytoplankton (type 1) sank to 300 m, indicating a higher turnover of salp pellets compared to krill FP in the mixed layer. Iversen et al.[31] documented similarly low export efficiencies of salp FP of ~13%. Direct flux estimates using gel

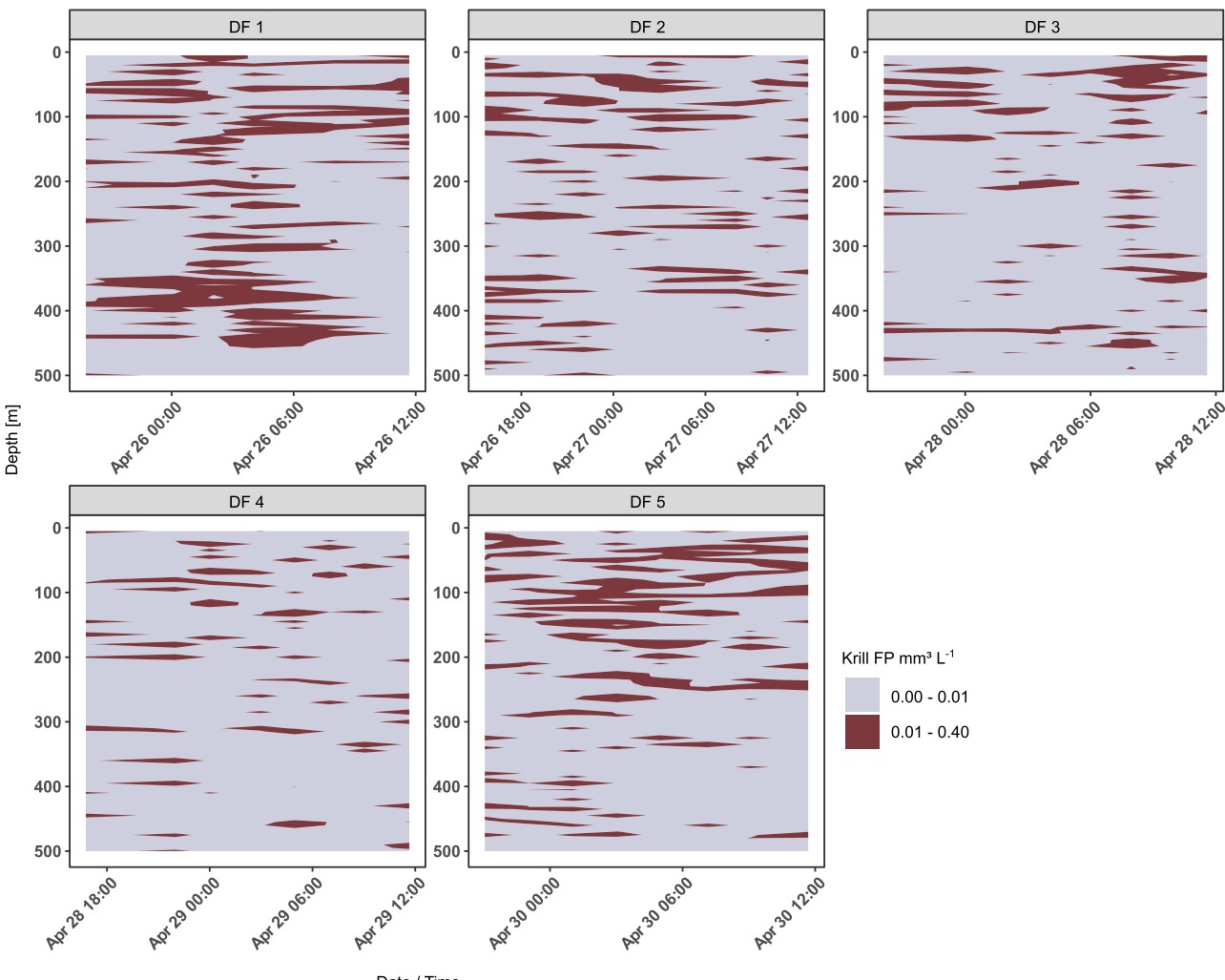

**Fig. 3 Volume concentration of krill faecal pellets (FP).** The volume concentration of krill FP is given in mm$^{-3}$ L$^{-1}$ as determined from the in-situ particle camera profile for each of the five drifting trap (DF) deployments, respectively. Clouds of krill FP sinking through the water column over time are shown on the x-axis for each DF deployment (DF 1–5).

traps indicate that the fragile nature of salp FP makes them more prone to fragmentation and recycling in the upper water column[31].

Carbon-specific microbial degradation rates were low for both salp and krill FP (0.3 vs. 1.2% d$^{-1}$, respectively) and were slightly lower than previous measurements at temperatures <4 °C[16,31,41]. We measured the oxygen consumption at temperatures between 1.5 and 4.5 °C. With a $Q_{10}$ factor of ~3.5 for aggregate degradation[41], the temperatures below 4 °C could explain the lower degradation rates compared to previous studies. Overall, with average sinking velocities of ~200 m d$^{-1}$ for krill FP and ~600 m d$^{-1}$ for salp FP, it is clear that microbial degradation alone cannot explain the 28% and 80% retention in the upper 300 m of krill and salp pellets, respectively. To assess other factors influencing the turnover of salp pellets besides microbial degradation, we tested for the impact of hydrodynamical stress (drag, friction) on the disintegration of sinking salp FP. For this, we conducted a roller tank experiment, where salp pellets were freely sinking at in-situ temperature for four days to imitate sinking from the surface to ~2.5 km depth. We followed the size-specific sinking velocity of the individual salp FP in the incubation using a video set-up method by Ploug et al.[42]. Throughout the four days of incubation, we did not observe any change in size-specific sinking velocity ($p = 0.21$, Chi-Sq. = 5.84, Df = 4, Kruskal-Wallis

rank sum test, Supplementary Fig. 8) or pellet volume ($p = 0.9$, $F = 0.04$, Df = 4, ANOVA).

Neither microbial degradation nor disintegration due to physical or hydrodynamical stress could explain the low export efficiency and high retention of salp FP in the mixed layer. Thus, we conclude that loosening and fragmentation of FP mediated by zooplankton activity (coprochaly, coprorhexy) were the main influencing factors, as also suggested by Iversen et al.[31]. This fragmentation leads to slower sinking velocities and provides a relatively larger surface area for microbial colonisation[16,43,44]. Accordingly, the physical structure of salp FP in the gel traps changed with increasing depth towards loose, fragmented and seemingly degraded salp FP, indicating they were modified during sinking (Fig. 4). Similarly, Iversen et al.[31] observed that loose, seemingly degraded pellets settled at about half the velocity of equally-sized, more densely packed, intact pellets. Such sinking velocities of ~150–200 m d$^{-1}$ are comparable to those of krill FP, indicating that even fragmented or partially degraded salp pellets still settle at high velocities and would thus contribute to the flux of carbon. We found that krill and both salp pellet types together accounted for ~75% of the total POC flux in sediment traps at 300 m depth. To estimate the potential contribution of fragmented salp FP to the carbon flux, we assume the remaining 25% of total carbon were in fact disintegrated parts of salp FP, which

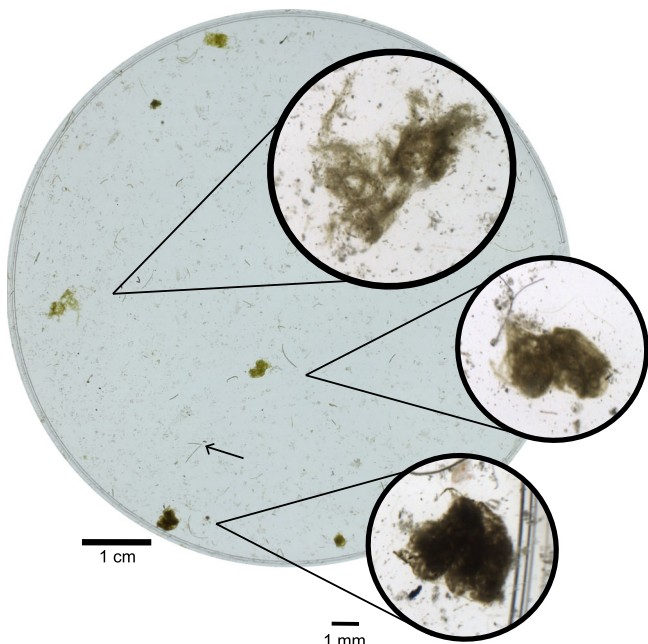

**Fig. 4 Different types of salp faecal pellets found in the gel inserts of a drifting sediment trap at 200 m.** The bottom crop picture shows a densely packed, intact salp faecal pellet (FP), likely including krill FP. The crop picture in the middle shows a more loosely packed salp FP, indicating that it was modified during sinking. The upper crop picture shows a very fragmented salp FP. The scale bar for the entire gel trap is 1 cm, the scale bar for the zoom-in crops is 1 mm. The black arrow indicates two krill FP for comparison.

we were not able to identify in the sediment traps or from the camera profiles. This would mean an additional average export flux of salp FP of 20 mg C m$^{-2}$ d$^{-1}$ to 300 m, which compared to the average flux of salp FP in the sediment traps at 300 m (28.4 mg C m$^{-2}$ d$^{-1}$) would increase salp FP export flux by ~70%. Based on on-board observations, we assume that salp FP consisting of phytoplankton (type 1) are generally more fragile than salp FP that contain krill FP (type 2) and are consequently fragmented at higher rates in the mixed layer. Applying a very rough estimate and assuming that the total share of the additional flux of fragmented salp pellets are type 1 pellets, this would result in a higher export efficiency of type 1 pellets of about 53% to 300 m. However, this would still mean that about half of the salp FP are retained in the mixed layer and highlights that the fate of salp FP needs to be further studied to better assess their role in the carbon cycle in the SO. Moreover, fragmented salp FP would be smaller and sink slower, which would allow more time for microbial degradation and ingestion of the FP fragments in the upper 300 m. This would explain why we did not observe recognisable fragments of salp FP in the sediment traps or in the camera profiles. However, we cannot conclusively determine the fate of salp FP fragments in this study.

Salps might also significantly contribute to the active carbon flux by performing diel vertical migrations to several hundred metres depth[40]. The production of FP below the mixed layer could significantly increase the export efficiency of carbon[7]. Global models indicate that this active transport of carbon by zooplankton could increase carbon flux estimates by 14%[9]. However, this contribution is difficult to quantify and needs to be elucidated in future studies. Another potential bias in the calculation of the export efficiency of salp FP may result from the different integration depths used for krill and salp FP due to the different methods applied to measure abundance (net hauls vs. hydroacoustics). Estimating the FP production by salps integrated

for the upper 200 m results in 70.14 mg C m$^{-2}$ d$^{-1}$ (vs. 59.62 mg C m$^{-2}$ d$^{-1}$ for the upper 170 m), which is 3% less than the export efficiency calculated based on the integrated FP production for the upper 170 m (20 ± 8.1%). Therefore, we assume that the resulting bias does not have a significant effect on the conclusions drawn.

Similar to the observed fragmentation of salp FP, we found a tendency towards more fragmented krill FP with increasing depth based on the vertical profiles conducted with the in-situ particle camera, indicating that krill pellets were modified by zooplankton (Supplementary Fig. 9). Copepods have been observed to fragment FP[44] and it was recently shown experimentally that, depending on the FP composition, POC turnover rates can be significantly higher in fragmented compared to non-fragmented FP[45]. On a larger scale, Briggs et al.[46] showed that fragmentation of sinking particles may explain 50% of the total carbon flux attenuation in the SO and North Atlantic. We therefore conclude that fragmentation of FP is a key parameter in controlling the flux of krill and salp faecal pellets at the AP. Particularly for salp FP, fragmentation seems to have been an important process that explains their high retention in the mixed layer. Although total carbon was higher in salp FP, krill FP had a higher carbon to volume ratio, making them a potentially more favourable food source for detrital and suspension feeders[47]. However, there are fundamental differences in the structure, packing and composition of krill and salp FP. Krill have a very effective digestion and are able to also grind larger prey items such as diatom frustules using the gastric mill in their digestive tract[48]. In addition, krill pellets are surrounded by a chitinous, peritrophic membrane, enabling dense packing and increasing pellet stability[49]. In contrast, salp FP lack such a membrane, which makes them more fragile, prone to fragmentation[50], and facilitates the colonisation by microbes. Moreover, salp FP often contain undigested food particles and high shares of diatom remains[36,39,51]. Thus, salp FP contain organic matter that might be more easily accessible than krill FP, particularly when pellets are broken into smaller pieces and subsequently sink at lower velocities. Fragmented FP can be colonised by protozooplankton (e.g. dinoflagellates, ciliates), which were shown to play a significant role in the degradation of FP[52]. Fragmentation of salp FP might also facilitate the release of micro- and macronutrients, such as iron, and ammonium, fuelling the lower food web. It was recently shown that iron released from salp FP might be more efficiently used by phytoplankton compared to iron released from krill FP[53]. Thus, grazing and remineralisation of salp FP might benefit the epipelagic food web in so far unaccounted manners, particularly when salps occur in high abundances during swarm formation and in years with outstandingly high abundances ('salp years')[54].

The observation of salps feeding on krill FP and the subsequent production of salp FP packed with krill FP (type 2 FP) has various implications for the carbon flux of both krill and salp pellets. The ingestion of krill FP by salps could increase the export efficiency of krill FP as they are packed into salp pellets with higher sinking velocities. Conversely, we have shown that salp pellets are exposed to higher turnover in the mixed layer, which could result in a lower export efficiency of krill FP. It is unclear whether salps mainly feed on phytoplankton at the surface and on krill FP at greater depth and at which depth the resulting FP are produced. We observed a high flux of type 2 salp FP during DF 1 (Table 4), which coincided with the occurrence of a krill swarm and subsequent high pellet production by krill. Overall, we showed that type 2 FP accounted for about 50% of the salp-mediated FP flux at 200 and 300 m in sediment traps. This coprophagous behaviour of salps was observed at several stations where krill and salps co-occurred. Hence, this might be a common phenomenon that will be of increasing importance with regard to the

increasingly overlapping distribution of krill and salps due to warming temperatures, and needs to be further elucidated. On a larger scale, this behaviour would have implications for the bio-geochemical cycling of krill FP. If salps feed on krill FP at depth and bring these pellets back to the surface, where they are egested again, krill FP would be exposed to a second degradation cycle. Besides the effects on the carbon cycle of krill FP, type 2 salp FP also contribute to the export of carbon. In this study, we did not include type 2 pellets in the calculation of the export efficiency of salp FP, due to the many unknown factors on the formation and flux of theses pellets. This might result in a potential underestimation of the export efficiency of salp FP and an overestimation of the export efficiency of krill FP, which is difficult to distinguish from the flux of krill FP and should be further investigated.

The average flux of salp FP in sediment traps was 29.7 mg C m$^{-2}$ d$^{-1}$ at 200 m, which is in accordance with previous studies (0–23.4 mg C m$^{-2}$ d$^{-1}$)[31,39]. The average contribution of salp FP to the total POC flux was 28.9%, which is higher than previously reported[31]. At the WAP, Gleiber et al.[15] found lower average contributions of salp pellets, but a similarly high variation (5–66%). Krill FP have been previously shown to drive high fluxes of POC particularly when forming large and dense swarms[16,18,34,38]. Similarly, we found that krill FP accounted for up to 92% of the total flux of POC and that krill FP were exported at high efficiencies to 300 m, agreeing with earlier studies showing export efficiencies of FP up to 100%[18]. This supports the notion that during the formation of vast swarms, krill can produce a heavy rain of FP that exceeds the capacities of detritivores and scavengers such as tunicates, pteropods, copepods or amphipods and leads to a disproportionally high flux of krill FP[11,14]. Together with recent findings on the combined impact of krill FP, carcasses and exuviae, which could account for 92% of the annual total flux of carbon[34], this highlights the exceptional role of krill in the biological carbon pump in the SO.

The potential daily FP flux by krill varied greatly over the course of five days ranging from 7 to 1895 mg C m$^{-2}$ d$^{-1}$, coinciding with the occurrence of vast swarms that reached abundances of up to 1000 Ind. m$^{-3}$, and which are in the same range as reported from other studies using hydroacoustics in the Scotia Sea[55]. These abundances contrast with the maximum observed abundances of salp swarms, which rarely exceed 10–30 Ind. m$^{-3}$[56,57]. The observed krill swarms stayed in the vicinity of the traps for ~30 min to 2 h. Swarm abundance and residence time were considered for the calculation of the potential flux of krill FP, thus we assume that our estimated FP production by krill is robust and does not present an overestimation. The patchy distribution and occurrence of vast, dense swarms as observed during this study is typical for Antarctic krill[17,58]. Similarly, salps can form large and dense swarms and their extreme patchiness was previously confirmed by consecutive ~1 km hauls showing abundances varying by up to two orders of magnitude[59]. However, in contrast to krill, to date there is no established method to estimate the abundance and vertical distribution of salps using hydroacoustic surveys and estimates rely on the use of large plankton nets. The different methods used to assess the abundance of krill and salps may have resulted in an underestimation of the salp abundances that could potentially result in a lower estimated salp FP production. However, we expect this underestimation to be minor, as salps do not show escape behaviour from plankton nets. Assuming we underestimated the salp abundance and subsequent FP production by 10%, the resulting export efficiency of salp FP to 300 m would be 18.2% instead of 19.9% (Supplementary Table 5) and would therefore not influence the main conclusions drawn here. If the underestimation of salp abundance and FP production was 50%, the resulting export efficiency would be considerably less (13.3%). Thus, even a 50%

underestimation, which is very unlikely, would still mean that salp pellets are efficiently turned over in the upper water column. Moreover, in this study, net hauls using a Multiple Rectangular Midwater Trawl (Multi-RMT) between the surface and 330 m depth confirmed that salps were performing diel vertical migration to the surface at night and to depth during day (Supplementary Fig. 6) and therefore confirm that we were able to catch the salps that migrated to the surface at night by sampling the upper 170 m.

The patchy distribution of both krill and salps exacerbates the estimation of their varying contribution to the carbon flux. The direct combination of hydroacoustics and/or net tows with the deployment of sediment traps and in-situ camera systems, as employed here, offers the possibility to overcome biases due to the temporal and spatial separation of flux and abundance estimates and is pivotal to capture these episodic flux events. The assessment of the true extent of the role of krill and salps in biogeochemical cycles at temporal and spatial scales is a pressing concern for research, as krill is particularly susceptible to the ongoing climatic changes and increasing fisheries pressure[12,60].

The Antarctic Peninsula differs from other regions of the SO, as it is one of the hotspots of krill biomass[61] and also harbours a high biomass of salps[30]. Therefore, the AP is one of the few regions were krill and salps co-occur. Moreover, the AP region is one of the most affected by global warming in the Southern Ocean as well as globally[20,22], where shifts in plankton and grazer communities have been observed over the past decades[30,62]. In other regions of the SO, where salps are already the dominant species and krill biomass is low, the relative contribution of salp FP to the export of carbon is expected to be higher than shown here for the AP region. Previous studies documented different types of salp FP (loose and compact)[26,31] that ranged in their fragility. Thus, it is possible that the high retention of salp FP may already occur in regions dominated by S. thompsoni.

The ongoing warming trend at the AP has led to increasing salp abundances and a southward shift of the krill population[23,24,32]. This shift in the grazer community has previously been hypothesised to increase the export flux of carbon[39,54]. In contrast, we show that the fragile nature and high retention of salp FP in the upper water column limits their contribution to carbon export. Furthermore, increased remineralisation of salp FP would provide an additional source of nutrients in the surface ocean, which has the potential to further alter SO food webs. In combination with the decreased carbon fluxes associated with declining krill stocks, our results indicate that the SO could become a less efficient carbon sink for anthropogenic $CO_2$ in the future. At the same time, this loss in the carbon flux could partly be compensated by increasing salp abundances and their high faecal pellet production rates.

## Methods

**Field work**. This study was conducted during the cruise PS112 with RV Polarstern off Elephant Island (61°08′07.7″S, 55°12′14.9″W) at the northern tip of the Antarctic Peninsula in April 2018. Specimens of Antarctic krill and salps were collected by oblique tows using IKMT (Isaacs-Kidd Midwater Trawl, 505 μm mesh size, 1.8 m$^2$) or RMTs (Rectangular Midwater Trawl, 320 μm mesh size). To determine the abundance of salps, quantitative oblique tows were conducted at a ship speed of 2 knots to 170 m depth or 20 m above the bottom, respectively (Supplementary Table 2). The vertical distribution of salps was determined using Multi-RMT trawls covering a depth range from 0 to 330 m. Samples were processed on board immediately after each catch and enumerated to species level. Random subsamples were taken when the catch volume was larger than 2 L. To account for the diel vertical migration of krill and salps, we defined day and night according to the local times of sunrise and sunset as the periods from 06:00 a.m. to 19:00 p.m., and from 19:00 p.m. to 06:00 a.m., respectively (UTC −03:00). Field work was conducted in compliance with national and international regulations and authorisation was granted by the German Environment Agency (Umweltbundesamt, UBA; reference II 2.8 – 94003-3/409). No ethical approval was necessary, as this study did not include vertebrate animals.

**Hydroacoustic survey.** The biomass and vertical distribution of krill was estimated based on acoustic data using an echo sounder (SIMRAD EK60). The echo sounder was calibrated at four frequencies (38, 70, 120 and 200 kHz) at two locations at the South Shetland Islands (Admiralty Bay, Half Moon Bay) at the beginning of the cruise. Acoustic data, including day and night surveys, were analysed using the Echoview software (v8.0), following existing protocols[63]. A stochastic distorted wave-born approximation model was used to calculate the target strength of krill[64–68]. Krill biomass was calculated in g m$^{-2}$ based on the acoustic backscatter attributed to krill at 120 kHz[69] using the length frequencies sampled during the cruise. For the calculation of the krill specific FP carbon flux, this biomass was converted into abundances integrated over depth (Ind. m$^{-2}$) using Siegel's season specific length-weight relationship for krill[37], following the equation

$$W = a * AT^b \tag{3}$$

where $W$ is the weight of krill in mg, AT is the length in mm and $a$ and $b$ are the regression coefficients[37]. For the calculation, we used a mean length of krill of 42.9 mm (total range 28–56 mm) obtained from ten net tows conducted in the study area, corresponding to a wet weight of 515.9 mg Ind.$^{-1}$, which was also used for the conversion of krill biomass to abundance (Ind. m$^{-2}$) and for the calculation of potential FP flux.

**Standing stock of chlorophyll *a* and particulate organic carbon.** Concentrations of chlorophyll *a* and particulate organic carbon (POC) in the upper 200 m of the water column were measured from water samples collected with a CTD rosette (Sea Bird Scientific SBE911plus, Carousel Water Sampler SBE 32) at three to five depths per station. Subsamples of 2 L from the CTD rosette were filtered for POC and chlorophyll respectively on pre-combusted GF/F filters using a peristaltic pump at 200 mbar and stored at –20 °C until further analyses in the home laboratory. Chlorophyll *a* was extracted using 90% acetone, incubated for 2 h at 4 °C and homogenised using glass beads (Precellys, Bertin Instruments, France). Subsequently, the homogenised filter-acetone mixture was centrifuged for 10 min at 4332 × $g$ with 5000 rpm. Fluorescence was measured (Turner Designs 10 AU) before and after acidification with 1 N HCl. For the processing of POC samples see paragraph on export carbon flux and elemental analysis of particulate organic carbon and nitrogen.

**Faecal pellet production experiments.** Faecal pellet production rates of krill and salps were measured from incubation experiments using freshly caught animals during day and night (Supplementary Table 3). Krill were placed individually in 4.5 L buckets filled with ambient, unfiltered seawater from the depth of maximum chlorophyll (~40 m) with a replicate quantity of ten (Supplementary Table 3). A mesh (2.5 mm diameter) was placed on the bottom of the bucket to prevent the krill from breaking the pellets (coprorhexy), or ingesting them (coprophagy). Salps were placed individually (solitary stages) or as short chains of three to eight individuals (aggregate stages) into 20 L buckets filled with water from the upper 5 m. Incubation containers of both animals were kept in darkness at 0.5 °C in a temperature-controlled room for 6–9 h for krill, and 8–12 h for salps. Pellet production of salps was assessed in 2-h intervals, while krill FP production was assessed once at the end of each incubation. The faecal pellets of both species were carefully removed from the incubation containers using a wide-bore pipette and faecal pellet size (length and diameter for krill pellets; length, width, and height for salp pellets) and sinking velocity was measured. Subsequently, faecal pellets were pooled per experimental replicate, filtered onto pre-combusted GF/F filters and frozen at –20 °C for later analyses of their carbon and nitrogen content.

**Drifting sediment traps.** To measure the in-situ carbon flux we deployed free drifting sediment traps five consecutive times for ~24 h each (Supplementary-Table 1). The traps drifted along the northern part of Elephant Island between 60°55.622′ to 60°59.765′ S and 054°37.844' to 055°09.133′ W. The drifting trap array consisted of three collection depths: 100, 200 and 300 m. At each depth, four collection cylinders (84.95 cm$^2$ collection area each) were deployed in a construction with gimbal mounts to ensure their vertical position during the deployment (KC Denmark A/S). One cylinder per depth was equipped with an insert containing a viscous gel (Tissue-Tek, O.C.T.$^{TM}$ COMPOUND, Sakura) to preserve the size and structure of the sinking particles[35,70]. All collection cylinders were filled with GF/F filtered seawater with additional salt to increase the salinity by four units prior to the deployment. The drifting trap array consisted of a surface buoy including an Iridium satellite sender that provided the trap positions every ten minutes with a two-minute resolution. Fourteen small buoys acted as wave breakers to reduce hydrodynamic effects and two benthos floats were attached for buoyancy. After recovering the drifting traps, particles were allowed to settle for ~12 h in the dark at 0.5 °C before the overlaying water was syphoned until about 200 ml remained, containing the collected material at the bottom of the collection cylinders. Subsequently, the collected material from each collection cylinder was carefully rinsed into individual sampling containers, fixed with mercury chloride and stored at 4 °C for later analyses of the biogeochemical flux. The gel trap inserts were gently removed from the collection cylinders and immediately imaged on board before they were frozen at −20 °C.

Krill and salp FP were identified from the gel trap images, and the pellet size (length ($l$) and diameter for krill pellets; length ($a$), width ($b$), and height ($c$) for salp pellets) was measured in ImageJ (v. 1.53a, NIH, USA). Pellet volume was calculated assuming a cylindrical shape for krill FP ($V_{cylinder}$; Eq. 4) and an ellipsoid shape for salp FP ($V_{ellipsoid}$; Eq. 5). This volume was then multiplied by the respective POC/volume ratio to obtain the species-specific FP flux in mg C m$^{-2}$ d$^{-1}$.

$$V_{cylinder} = h * \pi * r^2 \tag{4}$$

$$V_{ellipsoid} = 4/3 * \pi * a * b * c \tag{5}$$

where $r$ is the radius of the krill pellets.

**In-situ camera profiles and vertical size-distribution of krill pellets.** Every drifting trap deployment was accompanied by a series of in-situ camera profiles to determine the vertical particle-size distribution and abundance from the surface to 500 m depth. The parallel camera profiles were performed with a lowering speed of 0.3 m s$^{-1}$ directly after the deployment of each drifting trap and thereafter at high temporal resolution of 3–4 h in close proximity to the drifting trap position. The in-situ camera system consisted of an industrial camera (Basler) with removed infra-red filter and a fixed focal length lens of 16 mm (Edmund Optics). The setup was equipped with a DSPL battery (24 V, 38 Ah) and operated by a single board computer. An array of infra-red LEDs in front of the camera severed as light source. This provided shadow images of the particles with a frequency of two images per second. Each image from the in-situ camera system captured particles in a water volume of 20.46 cm$^3$ every 15 cm. The camera set-up was mounted onto a platform equipped with a CTD device (Seabird SBE19) containing an oxygen, a turbidity, and a fluorescence sensor. Overall, we conducted 38 deployments with the in-situ camera covering five consecutively deployed drifting traps over a period of five days (Supplementary Table 4).

The vertical size-distribution of krill faecal pellets was quantified from the in-situ camera deployments. We processed the images with the Imaging Processing Toolbox in Matlab (R2019a, The MathWorks, Inc., Natick, MA, USA) to characterise individual particles[71]. Each image was converted into grey scale and the background was removed by applying a threshold value based on the median pixel value for the whole image multiplied by a factor of 1.3. The pixel numbers for each projected particle area was determined, and converted into equivalent spherical diameter (ESD) using the pixels to mm ratio of 24[71]. Krill FP were identified by visual inspection, and length and width were measured using ImageJ (v. 1.53a, NIH, USA) and converted into volume assuming a cylindrical shape. The number of pictures with krill FP was subtracted from the total number of pictures captured during the down cast of the in-situ camera to obtain the concentration of krill faecal pellets.

**Sinking velocities, faecal pellet carbon content and oxygen measurements.** Size-specific sinking velocity of faecal pellets was measured in a vertical flow chamber[72,73] filled with GF/F filtered ambient seawater at in-situ temperature and salinity (~1 °C, 34 PSU). The flow chamber consisted of a 10 cm tall cylinder made from Plexiglas with an inner diameter of 5 cm. In the middle of the flow chamber, at a height of 5 cm, a mesh was stretched, which created a uniform laminar flow field in the upper half of the flow chamber when an upward flow was applied. Single faecal pellets were carefully placed in the upper part of the flow chamber and the upward flow was adjusted with a needle valve until it balanced the sinking velocity of the faecal pellet so it was suspended one diameter above the mesh. The faecal pellet sinking velocity was calculated by dividing the flow rate by the cross-sectional area of the flow chamber. The sinking velocity of each pellet was measured in triplicates before the pellet size was measured (length, width, and height for the ellipsoid salp pellets, and length and diameter for the cylindrical krill pellets) using a calibrated ocular microscope lens. Subsequently, faecal pellets were filtered on pre-combusted GF/F filters for analysis of POC and particulate organic nitrogen (PON) content. Based on the different geometric shapes of krill and salp FP, the FP volume was calculated assuming an ellipsoid shape for salp (Eq. 5), and a cylindrical shape for krill pellets (Eq. 4)[15,31]. In addition, the ESD was calculated allowing for a better comparison between both pellet types. Some bigger salp faecal pellets could not be measured in the flow chamber as they were sinking faster than the maximum velocity of ~800 m d$^{-1}$ that can be measured in our version of the flow chamber. Therefore, the sinking velocity of these pellets was measured in a sinking column filled with the same water as the flow chamber. The sinking column was insulated to control the temperature and covered on the top to prevent evaporation and ensure a constant temperature and salinity. Pellets were placed into the column individually and the sinking time per distance was measured.

Additionally, in the flow chamber we measured the oxygen gradient through the pellet-water interface using an oxygen microelectrode with a tip size of 10 μm in steps of 50 μm (OX-10 oxygen sensor, Unisense), calibrated at air saturation and anoxic conditions. During these measurements, the pellets were suspended as described above. Following a protocol by Ploug and Jørgensen[72] the oxygen gradient was measured on the downstream side of the pellet. All measurements in the flow chamber were only conducted during calm sea conditions to avoid any impact from ship movement.

**Export carbon flux and elemental analysis of particulate organic carbon and nitrogen**. Export carbon flux of dry mass, POC and PON was collected with free-drifting sediment traps at 100, 200 and 300 m. To determine the flux, 1/5 split of the material collected in one of the three trap cylinders without gel per depth was filtered on a pre-combusted and pre-weighted GF/F filter. Swimmers were removed from the split manually using a microscope before filtration. After filtration, the filters were dried for 24 h at 40 °C before determining dry weight and the POC and PON content.

All pre-combusted GF/F filters containing faecal pellets from the pellet production experiments, flow chamber measurements, water column samples for standing stock assessment, as well as bulk samples from the drifting traps were treated in the same way. Samples were fumed with concentrated hydrochloride acid (HCl, 37%) for 24 h and dried for 48 h at 40 °C before determining dry weight on a Mettler Toledo UMX 2 scale (0.1 µg sensitivity). The dried filters were then packed in tin cartridges and analysed on a EuroEA Elemental Analyser (precision of ± 0.7 µg or ± 0.3%). Blank filters were used to correct for any contaminations.

**Statistics**. To test for the differences in the flux of faecal pellet between krill and salps in the drifting traps across the different depth we used an analysis of variance (ANOVA) after validating the assumptions of homogeneity of variance and normal distribution. The differences in the total POC traps between the different traps was tested using a Kruskal-Wallis rank sum test. All analyses were performed in R version 3.6.1[74], plots were generated using the 'ggplot2' package (version 3.3.2) in R[75].

**Reporting summary**. Further information on research design is available in the Nature Research Reporting Summary linked to this article.

## Data availability
The carbon and abundance data generated in this study are provided in the Supplementary information.

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

## Acknowledgements

We are thankful to the captain, crew and the scientific staff of Polarstern cruise PS112 for their support with sampling and on-board measurements. Further, we would like to thank Larysa Pakhomova for helping with measurements of salps and FP production experiments. Thanks to Kerstin Oetjen for POC- and chlorophyll-, and Sandra Murawski for CN measurements. Thanks to Ryan Driscoll for helpful discussions on krill abundances and distribution. This study was part of the project "Population shift and eco-system response – krill vs. salps" funded by the Lower Saxony Ministry of Science and Culture (MWK) lead by B.M. M.H.I., C.M.F., C.K. and S.S. were supported by the HGF Young Investigator Group SeaPump "Seasonal and regional food web interactions with the biological pump", VH-NG-1000. CMF was additionally supported by the AWI Strategy Fund project EcoPump. MHI was additionally supported by the DFG Research Center of Excellence "The Ocean Floor – Earth's Uncharted Interface": EX-2077-390741603.

## Author contributions

N.C.P., E.A.P., M.H.I. and B.M. conceptualised the study. N.C.P., M.H.I., C.M.F., C.K., E.A.P., F.K. and B.M. performed fieldwork. F.K. provided the measurements of primary production. M.B. and A.S.B. calibrated the acoustic system and collected acoustic data on board of RV Polarstern. X.L.W. and J.C.Z. conducted hydroacoustic analyses and krill biomass estimates. S.S. and N.C.P. conducted work in the home laboratory. N.C.P. and M.H.I. performed data analyses. N.C.P. drafted the manuscript, with the help of all authors.

## Funding

## Competing interests

The authors declare no conflict of interest.

## Additional information

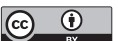

