## [Peer Review File · Nature Communications]

REVIEWER COMMENTS

Reviewer #1 (Remarks to the Author):

Review of the manuscript "Shifts in krill and salps alter the efficiency of the carbon export in the Southern Ocean" by Pauli et al.

The paper by Pauli et al. addresses how and why the observed shift in the Antarctic WAP zooplankton community from krill to more salps can impact the vertical carbon flux through the fate of the respective faecal pellets (FP), and how efficient they are exported. Both salp and krill FP are large and considered fast sinking, carbon rich and of great importance for the biological carbon pump. The paper challenges a suggested strengthening of the carbon pump with increased salp abundances based on investigations of fast sinking salp FP. Through a very extensive and well-designed field investigation using an impressive set of complementary approaches, the authors compare the salp- and krill FP production and flux, related to the actual abundances of the two species and measured vertical export. They measure the microbial degradation, the sinking rates and explore different types of FP produced based on salp trophic modes. They show that the story is not that simple, and that the efficiency of the carbon-pump may actually be weakened by a shift from krill to salps, even if the latter produce larger and faster sinking FPs.

The presented work is impressive and extensive in terms of complementary approaches that explore all or most aspects of the problem that is addressed. Despite being a field investigation, abundances, rates and properties of the important aspects has been measured. The data provided is of high quality, they are interpreted carefully to reveal the fate of krill and salp FP that are key to the carbon export in the Southern Ocean. The data and the interpretation provide solid support for the conclusions.

The methods are state of the art and how they are combined are novel. The combined approach results in a robust and unique dataset that provides new insight to complex processes needed to understand ecological responses with impact on biogeochemical cycling. The combination of acoustics for species abundance, species specific FP production measurements, size-specific FP sinking rates, FP microbial degradation rates and particle profiling, and a series of drifting sediment trap measurements, including use of gel to preserve the particles collected is rare. The combined approach provides a relatively complete picture of complex processes that reveals counterintuitive results and argues for a revision of the suggested hypothesis that an increased abundance of salps would increase the carbon flux. The impact includes insight to the complexity of properties that regulates the biogeochemical cycling. FP from larger organisms like krill are important vehicles for vertical flux in many regions, and this paper provides an approach to address the many aspects involved, and highlights key processes in FP retention and flux regulation. With the ongoing changes in climate that impact the zooplankton community composition globally and especially at high latitudes, this topic is of high general importance, and brings important attention to the many processes involved in and important for vertical carbon flux regulation.

Data and methodologies are appropriate, convincing and reproduceable through a well written Methods chapter. Methods are well described in text and with references. Results are generally well described in text, and through table and figures, both in the MS and supplementary, with some improvements suggested.

The analytical approach as a whole and across the different methods used are sound and reasonable both with respect to analysis and interpretations.

Suggested improvements

I miss a comment in the discussion on the investigated WAP region, relative to the Southern Ocean as a whole (addressed in the title). The different regions in the Southern Ocean have different relative abundance of krill and salps, with WAP being the region with most krill. The results can therefore not be directly

extrapolated to the remaining Southern Ocean, where salps already is the most important species.

It would also be good to see a sentence explaining why the type 2 FP from salps not are included in the summary of the efficiency of the FP flux, and if the use of different integration depths for salps and krill (170 m vs 200 m) impact the estimated efficiency.

I have provided some specific comments to the text given below, and there is a need to check numbers given in the text with the tables. They do not always match.

There are also some suggestions for consistency in use of station names (several versions).

Clarity and context

Most of the paper is well written and easily accessible. A few paragraphs are addressed in the suggested improvements and specific comments. The figures are informative and provide a solid documentation of the data, including pictures that are appreciated. The context is generally good, but as mentioned above, a comment on the WAP relative to the Southern Ocean – and the impact of regional differences would be reasonable given the title and conclusion including the Southern Ocean, and the investigation site being WAP.

References

The references are relevant and appropriate.

Specific comments

Effects for the southern ocean: Regionality vs WAP (heading vs data– extrapolation – and lack of a discussion related to the regionality and variability -i.e l 72 – ref to [23] Atkinson et al 2004, stating the WAP is the most krill intensive region, the others are more low prod and dominated by salps. A discussion on the relative importance of a similar transition for the WAP would at least acknowledge the regional variability, and that the effect cannot directly be scaled up to the entire Southern Ocean). Again, in the final paragraph of the discussion, it would be good to include some sentences on the justification of extrapolating the results from the WAP to the Southern Ocean, or the relative impact of the change going on at this site where the highest krill abundances are found.

Notion of stations – be consistent: The stations are referred to in many different ways that makes it harder to follow (DF 6-10 (Fig 1-3), Stn 118-122 (FigS1, S4, i.e text L327), dates (fig S7), last station (text), ..., and when some data is given for 5 stations and other for 6 – supplementary fig S7, it is challenging)

l 32 – suggest to specify: vertical carbon flux

l 56 – missing word -...varies ranges...

l 57 – “different regions” across the entire Southern Ocean? Or WAP neighbor region? Examples would help getting the larger context

L122-130: A comment on the different integration depths for krill (0-200 m) and salps (0-170 m) would be good, since the result in abundances m⁻², they are not directly comparable (would be more krill than salps). Does this difference also impact the potential flux estimates? They relate to sed traps at 200 and 300 m for both.

L179 (and suppl. Table 2): for the comparison of type1 vs type 2 FP carbon content, the st dev for both would be helpful. The Suppl. Table S2 does not provide any information on type 1 or 2 for the salp pellets. Given they are produced with surface water it is likely Type 1, but then info on Type 2 FPC data is missing.

In the paragraph on potential FP flux I struggle getting the clear overview, as well as matching the numbers and pellet types in the text and the tables:

L 189: Is the FP prod rates referred to Type 1?

L 192: would be helpful to write potential daily flux, to understand that the 59.62 mgm⁻²d⁻¹ is the sum of the day and night numbers given in the next line (and that could be made even clearer saying “...at night compared to day (39.78 vs 19.84 ..)”.

L193: balance in comments - the two-fold increased in salp FP at night is mentioned, but not the 30-fold

increase in krill FP at day (L205-206, although due to 1 station)

L 194- 197: this argument refers to Type 1 pellets – if potential flux for Type 2 pellets are not made, it could be good to just state that

L201: I don't understand the sentence "the resulting potential flux of FP showed large difference between traps ..."do you mean above traps or between trap stations? The traps contain FP flux, not potential flux

L203-206: it would be helpful if you state that the range of day comparison range includes DF1-6, while the range of night comparison include DF 1-5, and then DF 6 (that is extremely high), is given separately. I struggled to understand why the range of night was so much lower than the specified station.

L207: the number given – is that the average potential flux, or the potential flux for the station where primary production was measured?

L216-219: is the point here that the dense swarm at 300-500 m was 20 times the abundance estimated for the integration of 0-200 m? In that case, include "...average of 0-200 m standing stock (4076 ind...)

L220- 224: The text and numbers given here confuse me a bit. Again dense clouds to 250 m – meaning they are not included in the 0-200 m based abundance? The numbers in L221 – does not match Table 3. I don't find abundances of 195 232 ind m⁻² in Table 2 providing krill abundance data (55.63±21.69, and 189.66±182.42 ind m⁻²), the krill FP flux of 1895.3 in the text does not match table 3 (flux data range 20-42 mg C m⁻² d⁻¹ for DF10, but almost match the potential flux given as 1892.55 mg c m⁻² d⁻¹). L223- how can the low krill abundances referred to for drifting traps 8 and 9 results in a potential FP flux of 710.5 mg C m⁻² d⁻¹ (exceeding by far the potential flux (?) of 35.8 mg C m⁻²d⁻¹ (L219)? Check with corresponding tables, and the use of flux versus potential flux.

L248 (and Fig. 3)– It would be useful here with a sentence explaining why the type 2 FP is not included in the discussions of the export efficiency. I acknowledge they make a loop for the krill FP, that is hard to illustrate in Fig 3, and can be discussed with respect to direct transfer of phytoplankton-based carbon, they do export some of the krill FP that is retained down, so maybe a stippled line in the figure and a sentence in the discussion would acknowledge that there is a contribution although hard to quantify. I am aware it is addressed 320-336, but a clear motivation stating that they are not included in the efficiency budget, and how does that impact the efficiencies of krill and salps in the text at L 248 is appropriate.

L281: A sentence introducing that the following assumptions attempt to address a potential underestimation of the salp FP fluxes would be helpful. It would also be helpful to include how this assumption of the remaining POC being degraded salp FP would impact the estimated relative importance/ efficiency of krill vs salp FP export

L351-352: the numbers in the text does not match table 3 (3-1893 in table vs 7-1895 in text)

L371: is the krill threatened or exposed to...

L 743-744- Table 3: why is the potential salp FP flux given as mean values without SD, when the suppl. Table S2 refers to many experiments? Could the difference in integration depth (salps 0-170 m vs krill 0-200 m impact the estimated potential FP flux? Would a potential FP flux for salps increase with the abundance being integrated to 200 m instead of 170 m, and thus impact the flux efficiency, related to the 200 or 300 m trap flux that is fixed?

L761 – Fig 3: I don't understand where the 42% number in the legend come from, what does it mean? And why is it included in the text and not the figure. And could it be possible to indicate a loop or dashed track for the Type 2 pellets here?

Suppl. table 1 – would be great with station depths included. Not shown (as far as I see)

Suppl. Table 2 – is the mean size given the length, the width, or the ESD? Is the salp exp for Type 1 FP?

Why isn't there data on the carbon content of FP per volume for salp FPs?

Suppl. Fig s7: I struggle to understand what DF they refer to, as there were 5 deployments and are 6 plots here? Is it the right or the middle lower panel that describes the high krill densities observed at the "last DF station"? The middle has a scale to 10 000 ind m⁻², while the right panel show most points (at a scale to 4 ind m⁻²...).

Reviewer #2 (Remarks to the Author):

Nature Communications manuscript NCOMMS-21-10982

Review of Pauli et al: " Shifts in krill and salps alter the efficiency of carbon export in the Southern Ocean"

In this study the authors combine a multitude of measurements, such as of krill and salp abundance together with sediment traps and with experiments of fecal pellet production of both krill and salp. Among other things they measure the carbon content and microbial degradation of fecal pellets and present the many different results this experimental setup produces. A key result of this work is that while salps produce more fecal pellets that sink faster than krill fecal pellets, these fecal pellets seem to be retained longer in the upper 300m and hence contribute less to the carbon export to 300m. The authors speculate that declining krill stocks may make the Southern Ocean a less efficient CO₂ sink in the future ocean.

This study is quite extensive; utilizing and combining different approaches and extensive analysis, all of which are satisfyingly documented in the manuscript. The approach chosen is valid and robust, and a natural step forward after the 2017 Iverson et al. paper in DSRII found that although salps produced fast sinking fecal pellets these pellets were retained surprisingly long in the upper water column. They suggested break-up and loosening of the pellets by zooplankton as the main cause, which is also one major conclusion of the current paper.

The paper is well written, particularly the Introduction, however with progression of the paper it gets harder and harder to follow it, possibly because of the many results and conclusions that need to be covered. I recommend the following revisions of the paper:

I suggest to highlight better the novelty of this study. The authors mention in line 81 that "Studies directly comparing the contribution of salp and krill FP are lacking." But because the results of this study support/are similar to previous publications (e.g. Iverson et al., 2017 and others) it is easy to overlook this study is more complex etc. This should be stated ore clearly in the text.

My main concern with this study is that the abundance of salp in this study is measured very differently than that of krill: net hauls vs. acoustic measurements. While I understand why this is done, and it is well explained in lines 362 onwards, it is a big problem when comparing the results for krill and salp as it introduces biases, as the authors themselves mention. In a study like this I would expect a more differentiated discussion of this bias. How big of a bias is introduced here, what does that mean for your results? Could that lead to higher/lower estimates of salp concentrations and/or fecal pellet production? It will give you an uncertainty margin that should be defined. This point to me is one of the weak points of this study and should be much better discussed.

Further, the paper mentions repeatedly that krill cause extreme events and at other times are absent or abundance is low. And salp adapt quicker to environmental changes (food availability) and are a more continuous factor in the study area if I understand correctly. So what does that mean for the results presented here? This somehow ties into my comment above. Firstly, one would expect that salps would start to produce fecal pallets faster if food increases possibly changing the numbers calculated. Is it possible to assess this? Secondly, you are most likely not catching any very large aggregations of salp with the net hauls such as the very dense krill swarms which are detected by the acoustics. I'd like to see a better discussion of these points.

The title of the paper is not ideal. In this paper I could not find any discussion on what the shift of krill and salp expected in a future ocean actually means in numbers and in relation to the results presented. I can only find a general, final paragraph hypothesizing possible changes and unfortunately no calculation on how the efficiency will change in the future. Therefore, the title is misleading and should be changed.

I find there is some sloppy citing. While randomly checking some references, I found that in line 91 the citations 11 and 14 do not fit here. None of these papers claim krill FP are more efficient in carbon transport. You most likely mixed them up with other papers you reference. Please carefully check all your references.

Reviewer #3 (Remarks to the Author):

The manuscript presents the main findings of a study that assessed the relative contribution of krill and salp faecal pellets (FP) to vertical carbon flux during the cruise PS112 of the RV Polarstern off Elephant Island 383 (61°08'07.7"S, 55°12'14.9"W) conducted at the northern tip of the Antarctic Peninsula in April 2018. Results of the study indicate that on average, salps produced four-fold more FP carbon compared to krill, but the FP from both species contributed equally to the carbon flux to 300 m. The faecal pellets of the krill were efficiently exported to 300 m, while the fragile salp FP had strong retention due to fragmentation in the mixed layer. Decreased carbon fluxes associated with declining krill stocks and retention of salp FP indicate that the Southern Ocean could become a less efficient carbon sink for anthropogenic CO₂ in the future.

Overall, I found the manuscript to be well written and easy to read. Below I have listed a number of comments/suggestions that the authors should consider before the manuscript can be considered for publication.

1. I would encourage the authors to revise their abstract to include some actual data. Also, it would be useful if the authors could include information on where and when the study was conducted.
2. Line 48. Grazing by zooplankton may increase vertical carbon flux through diel vertical migrations and their grazing activities which produce faecal pellets which sink to depth.
3. Line 56. The sentence is grammatically incorrect and needs to be revised.
4. Line 67. Please check if the actual increase in seawater temperature increase 60C.
5. Given the importance of phytoplankton size in mediating the partitioning of carbon between krill and salps, do the authors have any data on the size structure of the phytoplankton community during the period of study?
6. It is likely that the net tows conducted only to a depth of 170m during the daytime sampling would have substantially underestimated the krill abundance and biomass. Indeed, it is worth noting that the discrepancy in krill abundances and biomass during the daytime and night-time samples is highlighted in Table 2. How might this underestimation have impacted the overall results of the investigation?
7. I would encourage the authors to include the range of values for the different measurements in the results section rather than a single value.
8. It is evident from Table 1 that only a single estimate of primary production was conducted during the survey. Given the inherent variability in daily primary production rates in the Southern Ocean, are the authors convinced that their estimates of the percentage of primary production exported by the salps and krill to depth are realistic?
9. Line 425. Faecal pellet production rates of krill and salps were measured from incubation experiments using freshly caught animals. Please clarify if these samples were collected during the daytime/night time since this may have influenced the amount of pigment in the guts of the krill.
10. Line 507. The pellet volume was calculated assuming an ellipsoid shape for salps, and a cylindrical shape for krill pellets. Please clarify or include a reference.
11. Figure 4 of the manuscript is redundant and can be omitted from the manuscript.

Response to reviewer comments on manuscript NCOMMS-21-10982 by Pauli et al.

Reviewer #1 (Remarks to the Author):

Review of the manuscript “Shifts in krill and salps alter the efficiency of the carbon export in the Southern Ocean” by Pauli et al.

The paper by Pauli et al. addresses how and why the observed shift in the Antarctic WAP zooplankton community from krill to more salps can impact the vertical carbon flux through the fate of the respective faecal pellets (FP), and how efficient they are exported. Both salp and krill FP are large and considered fast sinking, carbon rich and of great importance for the biological carbon pump. The paper challenges a suggested strengthening of the carbon pump with increased salp abundances based on investigations of fast sinking salp FP. Through a very extensive and well-designed field investigation using an impressive set of complementary approaches, the authors compare the salp- and krill FP production and flux, related to the actual abundances of the two species and measured vertical export. They measure the microbial degradation, the sinking rates and explore different types of FP produced based on salp trophic modes. They show that the story is not that simple, and that the efficiency of the carbon-pump may actually be weakened by a shift from krill to salps, even if the latter produce larger and faster sinking FPs.

The presented work is impressive and extensive in terms of complementary approaches that explore all or most aspects of the problem that is addressed. Despite being a field investigation, abundances, rates and properties of the important aspects has been measured. The data provided is of high quality, they are interpreted carefully to reveal the fate of krill and salp FP that are key to the carbon export in the Southern Ocean. The data and the interpretation provide solid support for the conclusions.

The methods are state of the art and how they are combined are novel. The combined approach results in a robust and unique dataset that provides new insight to complex processes needed to understand ecological responses with impact on biogeochemical cycling. The combination of acoustics for species abundance, species specific FP production measurements, size-specific FP sinking rates, FP microbial degradation rates and particle profiling, and a series of drifting sediment trap measurements, including use of gel to preserve the particles collected is rare. The combined approach provides a relatively complete picture of complex processes that reveals counterintuitive results and argues for a revision of the suggested hypothesis that an increased abundance of salps would increase the carbon flux. The impact includes insight to the complexity of properties that regulates the biogeochemical cycling. FP from larger organisms like krill are important vehicles for vertical flux in many regions, and this paper provides an approach to address the many aspects involved, and highlights key processes in FP retention and flux regulation. With the ongoing changes in climate that impact the zooplankton community composition globally and especially at high latitudes, this topic is of high general importance, and brings important attention to the many processes involved in and important for vertical carbon flux regulation.

Data and methodologies are appropriate, convincing and reproducible through a well written Methods chapter. Methods are well described in text and with references. Results are generally well described in text, and through table and figures, both in the MS and supplementary, with some improvements suggested. The analytical approach as a whole and across the different methods used are sound and reasonable both with respect to analysis and interpretations.

We thank the reviewer for the valuable feedback, which has helped to clarify and improved the manuscript. Please see our detailed responses to the specific comments below.

Suggested improvements

Comment 1.1: I miss a comment in the discussion on the investigated WAP region, relative to the Southern Ocean as a whole (addressed in the title). The different regions in the Southern Ocean have different relative abundance of krill and salps, with WAP being the region with most krill. The results can therefore not be directly extrapolated to the remaining Southern Ocean, where salps already is the most important species.

Response: We agree with the reviewer and have added a paragraph on the distinctiveness of the WAP region to the discussion section:

“The Antarctic Peninsula differs from other regions of the SO, as it is one of the hotspots of krill biomass⁶¹ and also harbours a high biomass of salps³⁰. Therefore, the AP is one of the few regions where krill and salps co-occur. Moreover, the AP region is one of the most affected by global warming in the Southern Ocean as well as globally^{20,22}, where shifts in plankton and grazer communities have been observed over the past decades^{30,62}. In other regions of the SO, where salps are already the dominant species and krill biomass is low, the relative contribution of salp FP to the export of carbon is expected to be higher than shown here for the AP region. Previous studies documented different types of salp FP (loose and compact)^{26,31} that ranged in their fragility. Thus, it is possible that the high retention of salp FP may already occur in regions dominated by *S. thompsoni*.” (Lines 417-426)

In addition, also in response to a comment by reviewer #2, we have changed to title to emphasize the regional aspect of our study: “Krill and salp faecal pellets contribute equally to the carbon flux at the Antarctic Peninsula”.

Comment 1.2: It would also be good to see a sentence explaining why the type 2 FP from salps not are included in the summary of the efficiency of the FP flux, and if the use of different integration depths for salps and krill (170 m vs 200 m) impact the estimated efficiency.

Response: We agree with the reviewer and have added a paragraph explaining why type 2 salp FP were not included in the calculation to the results section:

“Type 2 salp FP were not included in the export efficiency calculation as we assume that they result from salps feeding on krill FP during daytime at greater depth and are excreted at night when salps migrate to the surface to feed on phytoplankton, based on a gut passage time of about 8 hours³⁸. This pattern might decrease the efficiency of carbon export by krill FP. However, currently we cannot provide an accurate estimate of the impact of this coprophagous feeding behaviour is on the efficiency of the biological carbon pump.” (Lines 204-209).

In addition, we have added a sentence to the discussion:

“Besides the effects on the carbon cycle of krill FP, type 2 salp FP also contribute to the export of carbon. In this study, we did not include type 2 pellets in the calculation of the export efficiency of salp FP, due to the many unknown factors on the formation and flux of these pellets. This might result in a potential underestimation of the export efficiency of salp FP and an overestimation of the export efficiency of krill FP, which is difficult to distinguish from the flux of krill FP and should be further investigated.” (Lines 369-374)

To account for the potential bias introduced by the different integration depths for krill and salp abundances (200 vs. 170 m) we have added a calculation estimating the FP production by salps integrated for 200 m and have added a paragraph to the discussion section:

“Another potential bias in the calculation of the export efficiency of salp FP may result from the different integration depths used for krill and salp FP due to the different methods applied to measure abundance (net hauls vs. hydroacoustics). Estimating the FP production by salps integrated

for the upper 200 m results in $70.14 \text{ mg C m}^{-2} \text{ d}^{-1}$ (vs. $59.62 \text{ mg C m}^{-2} \text{ d}^{-1}$ for the upper 170 m), which is 3% less than the export efficiency calculated based on the integrated FP production for the upper 170 m ($20 \pm 8.1\%$). Therefore, we assume that the resulting bias does not have a significant effect on the conclusions drawn.” (Lines 317-323)

I have provided some specific comments to the text given below, and there is a need to check numbers given in the text with the tables. They do not always match. There are also some suggestions for consistency in use of station names (several versions).

We have double-checked all numbers in text, tables, and figures and have edited or explained discrepancies where necessary. In addition, we have edited the station and drifting trap labels for consistency. Please see our responses to the specific comments below.

Clarity and context: Most of the paper is well written and easily accessible. A few paragraphs are addressed in the suggested improvements and specific comments. The figures are informative and provide a solid documentation of the data, including pictures that are appreciated. The context is generally good, but as mentioned above, a comment on the WAP relative to the Southern Ocean – and the impact of regional differences would be reasonable given the title and conclusion including the Southern Ocean, and the investigation site being WAP.

We have adjusted the title of the manuscript to make it clear that the present study was conducted at the Antarctic Peninsula in distinction to the entire Southern Ocean:

“Krill and salp faecal pellets contribute equally to the carbon flux at the Antarctic Peninsula”

Moreover, we have added a paragraph to the discussion section explaining the presented results in the context of the Southern Ocean in relation to the Antarctic Peninsula. Please see our response to comment 1.1 for more details.

References: The references are relevant and appropriate.

Specific comments

Comment 1.3: Effects for the southern ocean: Regionality vs WAP (heading vs data– extrapolation – and lack of a discussion related to the regionality and variability -i.e l 72 – ref to [23] Atkinson et al 2004, stating the WAP is the most krill intensive region, the others are more low prod and dominated by salps. A discussion on the relative importance of a similar transition for the WAP would at least acknowledge the regional variability, and that the effect cannot directly be scaled up to the entire Southern Ocean). Again, in the final paragraph of the discussion, it would be good to include some sentences on the justification of extrapolating the results from the WAP to the Southern Ocean, or the relative impact of the change going on at this site where the highest krill abundances are found.

Response: We have added a respective paragraph to the discussion section. Please see our response to comment 1.1 above.

Comment 1.4: Notion of stations – be consistent: The stations are referred to in many different ways that makes it harder to follow (DF 6-10 (Fig 1-3), Stn 118-122 (FigS1, S4, i.e text L327), dates (fig S7), last station (text), ..., and when some data is given for 5 stations and other for 6 – supplementary fig S7, it is challenging)

Response: We thank the reviewer for pointing this out and agree that the use of different station names was confusing. We have adjusted the labels for drifting traps and stations for more consistency and clarity throughout tables, figures and manuscript text.

Comment 1.5: l 32 – suggest to specify: vertical carbon flux

Response: Changed as suggested. (Line 27)

Comment 1.6: l 56 – missing word -...varies ranges...

Response: Changed to: "...the carbon flux varies and ranges up to.." (Line 54)

Comment 1.7: l 57 – “different regions” across the entire Southern Ocean? Or WAP neighbor region? Examples would help getting the larger context

Response: The sentence was changed to include the specific regions to which the value refers: “At the Western Antarctic Peninsula (WAP) and the marginal ice zone krill FP were shown to account for 17–72% of the total carbon flux^{14,15}.” (Lines 55-56)

Comment 1.8: L122-130: A comment on the different integration depths for krill (0-200 m) and salps (0-170 m) would be good, since the result in abundances m⁻², they are not directly comparable (would be more krill than salps). Does this difference also impact the potential flux estimates? They relate to sed traps at 200 and 300 m for both.

Response: We agree with the reviewer and have now included an additional calculation of the potential FP production of salps integrated for 0-200 m, which we discuss in a separate paragraph (Lines 317-323). The export efficiency estimates relate to the 300m depth only. Please also see our response to comment 1.2 above for more details.

Comment 1.9: L179 (and suppl. Table 2): for the comparison of type1 vs type 2 FP carbon content, the st dev for both would be helpful. The Suppl. Table S2 does not provide any information on type 1 or 2 for the salp pellets. Given they are produced with surface water it is likely Type 1, but then info on Type 2 FPC data is missing.

Response: The standard deviation for both values is now shown (Lines 174-175). In accordance with a comment by reviewer #3, we have adjusted the results section to include the data range or mean with standard deviation for all reported values.

In addition, we have added a sentence to the caption of Supplementary Table 2 (now Suppl. Table 3) to explain that the salp FP data refer to Type 1 FP. As the reviewer pointed out correctly, the FP produced by salps in the incubation experiments were Type 1 pellets, as they were produced based on in situ surface water. Type 2 pellets were mainly observed in the gel traps.

“Salp FP in the experiments were produced by freshly caught salps, which were conducted using surface in situ water. Thus, the pellets referred to in this table are Type 1 FP, as type 2 salp FP were mainly observed from gel traps.” (Supplementary Table 3)

In the paragraph on potential FP flux I struggle getting the clear overview, as well as matching the numbers and pellet types in the text and the tables:

Response: The paragraph was edited for clarity. Please see our responses to the specific comments below.

Comment 1.10: L 189: Is the FP prod rates referred to Type 1?

Response: The FP production rates refer to the incubation experiments performed on board. These were carried out using water from the chlorophyll maximum layer and therefore represent Type 1 pellets produced on a phytoplankton diet. The respective sentence was edited for clarity:

“...with the size-specific FP carbon production rates for Type 1 pellets (mg C Ind.⁻¹ h⁻¹, Eq. 2)...” (Line 195)

Comment 1.11: L 192: would be helpful to write potential daily flux, to understand that the 59.62 mgm-2d-1 is the sum of the day and night numbers given in the next line (and that could be made even clearer saying "...at night compared to day (39.78 vs 19.84 ..)").

Response: Edited as suggested: "The potential daily flux of Type 1 salp FP (phytoplankton) was $59.62 \pm 14.1 \text{ mg C m}^{-2} \text{ d}^{-1}$ (Figure 2, Table 3), with a 2-fold higher pellet production at night compared to daytime (39.78 ± 11.5 vs. $19.84 \pm 29.8 \text{ mg C m}^{-2} \text{ d}^{-1}$)." (Lines 198-201)

Comment 1.12: L193: balance in comments - the two-fold increased in salp FP at night is mentioned, but not the 30-fold increase in krill FP at day (L205-206), although due to 1 station

Response: We agree with the reviewer and have adjusted the respective section. We now report the mean FP production at night and during day with standard deviation. We have now clarified that these average values do not include the last station, where the FP production was disproportionately high ($1,892 \text{ mgCm}^{-2}\text{d}^{-1}$).

"The potential FP flux of krill in the upper 200 m during the deployments of drifting traps 1 to 4 showed no significant difference between night ($13.02 \pm 13.27 \text{ mg C m}^{-2} \text{ d}^{-1}$) and daytime ($8.54 \pm 10.51 \text{ mg C m}^{-2} \text{ d}^{-1}$). During the deployment of DF 5, the presence of a vast krill swarm resulted in a potential FP flux of $1,892.6 \text{ mg C m}^{-2} \text{ d}^{-1}$ during day and $2.8 \text{ mg C m}^{-2} \text{ d}^{-1}$ at night." (Lines 216-220).

Comment 1.13: L 194- 197: this argument refers to Type 1 pellets – if potential flux for Type 2 pellets are not made, it could be good to just state that

Response: As suggested, we have added a sentence explaining that the export efficiency calculation was not conducted for type 2 salp FP. Please see our response to comment 1.2 above for more details.

Comment 1.14: L201: I don't understand the sentence "the resulting potential flux of FP showed large difference between traps ..."do you mean above traps or between trap stations? The traps contain FP flux, not potential flux

Response: We thank the reviewer for pointing this out. The word "potential" was removed and the sentence was edited accordingly: "The flux of krill FP showed large differences between the single trap deployments, as well as between the three deployment depths (100, 200, and 300 m; Table 3)..."(Lines 212-214)

Comment 1.15: L203-206: it would be helpful if you state that the range of day comparison range includes DF1-6, while the range of night comparison include DF 1-5, and then DF 6 (that is extremely high), is given separately. I struggled to understand why the range of night was so much lower than the specified station.

Response: We have edited the respective paragraph for clarity to state that the mean values do not include the last station with extremely high FP flux. Please see our response to comment 1.12 above.

Comment 1.16: L207: the number given – is that the average potential flux, or the potential flux for the station where primary production was measured?

Response: The number given is the average potential flux based on the available primary production value. This information was added to the sentence for clarity: "On average, the potential FP flux of krill and salps accounted for 48.3% and 57.8% of the primary production, respectively." (Line 221)

Comment 1.17: L216-219: is the point here that the dense swarm at 300-500 m was 20 times the abundance estimated for the integration of 0-200 m? In that case, include "...average of 0-200 m standing stock (4076 ind...)

Response: We have rephrased this paragraph to better explain the numbers used. The "dense clouds" here refer to the krill FP identified from the in situ particle camera. The respective sentence was rephrased for clarity. Please also see our response to the comment below.

"The camera profiles revealed dense clouds of krill FP between 300 and 500 m depth during the first trap deployment (DF 1), which correlated to a high krill abundance in the top 200 m and a high potential krill FP flux (505 Ind. m⁻², 35.8 mg C m⁻² d⁻¹; Table 3)." (Lines 230-233)

Comment 1.18: L220- 224: The text and numbers given here confuse me a bit. Again dense clouds to 250 m – meaning they are not included in the 0-200 m based abundance?

Response: The 'dense clouds' described in this paragraph refer to the krill FP detected by the in situ particle camera, which we then correlated to the measured krill abundance that we measured in the top 200 m. We have rephrased the entire paragraph for more clarification:

"During the five drifting trap deployments, we deployed an in-situ camera system near the trap position to quantify the vertical abundance and size-distribution of sinking particles in approximately 4-hour intervals. This resulted in a total of 38 profiles with a vertical resolution of 15 cm. We identified the volume concentration of krill FP from the camera profiles, which supported the patchy vertical and temporal distribution of krill FP that was observed from both the sediment traps and the calculated potential FP flux (Figure 3). The camera profiles revealed dense clouds of krill FP between 300 and 500 m depth during the first trap deployment (DF 1), which correlated to a high krill abundance in the top 200 m and a high potential krill FP flux (505 Ind. m⁻², 35.8 mg C m⁻² d⁻¹; Table 3). Similarly, dense clouds were observed between the surface and 250 m during the last trap deployment (DF 5), corresponding to the occurrence of a dense krill swarm during daytime and a high potential flux of krill FP (3117 Ind. m⁻², 1892.6 mg C m⁻² d⁻¹; Table 3). Low in-situ concentrations of krill FP were observed during the deployments of drifting traps 3 and 4, where the krill standing stock at 0–200 m depth and thus the potential FP flux were low (51 to 135 Ind. m⁻²; 7 to 10.5 mg C m⁻² d⁻¹; Table 3)." (Lines 225-238)

The numbers in L221 – does not match Table 3. I don't find abundances of 195 232 ind m⁻² in Table 2 providing krill abundance data (55.63±21.69, and 189.66±182.42 ind m⁻²), the krill FP flux of 1895.3 in the text does not match table 3 (flux data range 20-42 mg C m⁻² d⁻¹ for DF10, but almost match the potential flux given as 1892.55 mg c m⁻² d⁻¹).

Response: We thank the reviewer for pointing out this discrepancy. We have adjusted Table 2 and 3 accordingly. The abundances for krill and salps are now shown in Table 3. For krill, we have included the mean abundances for each drifting trap deployment for day and night, respectively. In addition, we have adjusted the numbers in the text to be transparent with the table. In the previous version of the manuscript, we had used the maximum observed krill abundances for the respective stations in this paragraph. We agree that these numbers were confusing and we have edited the paragraph for more clarity. Please see the copy pasted part in our response to comment 1.18 above.

The particular krill FP flux of 1892.55 mg c m⁻² d⁻¹ is the potential FP flux during day, as now shown in Table 3 for DF 5. We have edited the respective sentence for more clarity: "...corresponding to the occurrence of a dense krill swarm during daytime and a high potential flux of krill FP (3117 Ind. m⁻², 1892.6 mg C m⁻² d⁻¹; Table 3)." (Lines 234-236)

L223- how can the low krill abundances referred to for drifting traps 8 and 9 results in a potential FP flux of 710.5 mg C m⁻² d⁻¹ (exceeding by far the potential flux (?) of 35.8 mg C m⁻²d⁻¹ (L219)? Check with corresponding tables, and the use of flux versus potential flux.

Response: We are afraid a formatting error occurred here. The resulting potential flux is not 710.5, but 7 to 10.5 mg C m⁻² d⁻¹. The respective part was edited: “Low in-situ concentrations of krill FP were observed during the deployments of drifting traps 3 and 4, where the krill standing stock at 0–200 m depth and thus the potential FP flux were low (51 to 135 Ind. m⁻²; 7 to 10.5 mg C m⁻² d⁻¹; Table 3).” (Lines 236-238)

Comment 1.19: L248 (and Fig. 3)– It would be useful here with a sentence explaining why the type 2 FP is not included in the discussions of the export efficiency. I acknowledge they make a loop for the krill FP, that is hard to illustrate in Fig 3, and can be discussed with respect to direct transfer of phytoplankton-based carbon, they do export some of the krill FP that is retained down, so maybe a stippled line in the figure and a sentence in the discussion would acknowledge that there is a contribution although hard to quantify. I am aware it is addressed 320-336, but a clear motivation stating that they are not included in the efficiency budget, and how does that impact the efficiencies of krill and salps in the text at L 248 is appropriate.

Response: We agree with the reviewer and have added a paragraph discussing the missing export efficiency calculation for type 2 salp FP:

“Besides the effects on the carbon cycle of krill FP, type 2 salp FP also contribute to the export of carbon. In this study, we did not include type 2 pellets in the calculation of the export efficiency of salp FP, due to the many unknown factors on the formation and flux of these pellets. This might result in a potential underestimation of the export efficiency of salp FP and an overestimation of the export efficiency of krill FP, which is difficult to distinguish from the flux of krill FP and should be further investigated.” (Lines 369-374)

Comment 1.20: L281: A sentence introducing that the following assumptions attempt to address a potential underestimation of the salp FP fluxes would be helpful. It would also be helpful to include how this assumption of the remaining POC being degraded salp FP would impact the estimated relative importance/ efficiency of krill vs salp FP export

Response: The respective sentence was rephrased accordingly:

“To estimate the potential contribution of fragmented salp FP to the carbon flux, we assume the remaining 25% of total carbon were in fact disintegrated parts of salp FP, which we were not able to identify in the sediment traps or from the camera profiles. This would mean an additional average export flux of salp FP of 20 mg C m⁻² d⁻¹ to 300 m, which compared to the average flux of salp FP in the sediment traps at 300 m (28.42 mg C m⁻² d⁻¹) would increase salp FP export flux by ~70%. Based on on-board observations, we assume that salp FP consisting of phytoplankton (type 1) are generally more fragile than salp FP that contain krill FP (type 2) and are consequently fragmented at higher rates in the mixed layer. Applying a very rough estimate and assuming that the total share of the additional flux of fragmented salp pellets are type 1 pellets, this would result in a higher export efficiency of type 1 pellets of about 53% to 300 m. However, this would still mean that about half of the salp FP are retained in the mixed layer and highlights that the fate of salp FP needs to be further studied to better assess their role in the carbon cycle in the SO. Moreover, fragmented salp FP would be smaller and sink slower, which would allow more time for microbial degradation and ingestion of the FP fragments in the upper 300 m. This would explain why we did not observe recognizable fragments of salp FP in the sediment traps or in the camera profiles. However, we cannot conclusively determine the fate of salp FP fragments in this study.” (Lines 295-311)

Comment 1.21: L351-352: the numbers in the text does not match table 3 (3-1893 in table vs 7-1895 in text)

Response: We thank the reviewer for pointing this out. The potential FP flux of krill was 1893 during daytime and 1895 mg C m⁻² d⁻¹ for 24 hours (1892.5 + 2.7). We have added a row for the total daily potential FP flux of krill and salps to Table 3 to make this more transparent.

Table 3:

Abundance [Ind. m ⁻²]		DF 01	DF 02	DF 03	DF 04	DF 05	Mean	SD	
Krill	Day	29.20	81.60	51.50	60.20	3117.00	667.90	± 1369.22	
	Night	505.00	175.00	76.90	135.00	56.40	189.66	± 182.42	
Salps	Day						93.9	± 149.3	
	Night						278.9	± 87.7	
Flux [mg C m ⁻² d ⁻¹]	FP Type	Depth [m]							
Trap flux									
Total POC flux		100	90.04	98.85	47.26	50.44	58.78	69.07	± 23.74
		200	199.93	67.55	59.79	54.42	169.27	110.19	± 68.95
		300	100.68	67.77	64.00	56.62	83.50	74.51	± 17.62
Salp FP flux	1	100	2.50	17.99	14.01	1.00	2.58	7.62	± 7.81
			0	11.02	2.85	2.94	2.61	3.88	± 4.17
	2	100	27.32	14.64	26.56	0.78	8.41	15.54	± 11.51
			36.34	5.69	20.32	2.24	5.94	14.10	± 14.24
	1	200	12.32	5.65	19.02	12.45	10.16	11.92	± 4.82
			33.67	6.82	11.99	21.75	8.20	16.49	± 11.24
Krill FP flux		100	35.05	16.19	25.26	26.76	28.85	26.42	± 6.83
		200	8.47	17.70	11.68	50.23	42.36	26.09	± 18.95
		300	21.69	53.24	20.62	13.80	20.16	25.90	± 15.59
Potential FP flux									
Salps – Day		>170					19.84	± 29.82	
Salps – Night		>170					39.78	± 11.52	
Salps – 24 h							59.62	± 14.09	
Krill – Day		>200	3.14	24.29	3.10	3.62	1892.55	385.34	± 842.61
Krill – Night		>200	32.71	8.58	3.91	6.87	2.76	10.97	± 12.37
Krill – 24 h			35.85	32.87	7.01	10.49	1895.31	396.30	± 838.07

Comment 1.22: L371: is the krill threatened or exposed to...

Response: the respective sentence was rephrased: "...as krill is particularly susceptible to the ongoing climatic changes and increasing fisheries pressure^{12,60}." (Line 415-416)

Comment 1.23: L 743-744- Table 3: why is the potential salp FP flux given as mean values without SD, when the suppl. Table S2 refers to many experiments?

Could the difference in integration depth (salps 0-170 m vs krill 0-200 m impact the estimated potential FP flux? Would a potential FP flux for salps increase with the abundance being integrated to 200 m instead of 170 m, and thus impact the flux efficiency, related to the 200 or 300 m trap flux that is fixed?

Response: Table 3 and Supplementary Table 2 are not directly related. While supplementary Table 2 (now Suppl. Table 3) refers to the on-board experiments only, Table 3 shows the results of the calculated potential flux based on the in situ abundances of krill and salps. For the calculation in the original manuscript version we used the mean salp abundance for 6 stations (three per day and night

respectively) and used this mean for the calculation of the potential flux, therefore no SD was available. We have now adjusted this calculation to account for every single station separately and then took the mean including a SD which is now included in Table 3 (please see our response to comment 1.21 above).

In addition, we have included a calculation on the potential FP production of salps integrated for 0-200m to be better comparable with the potential FP production of krill and to discuss the potential bias in the estimated flux. Please see our response to comment 1.2 for more details.

Comment 1.24: L761 – Fig 3: I don't understand where the 42% number in the legend come from, what does it mean? And why is it included in the text and not the figure. And could it be possible to indicate a loop or dashed track for the Type 2 pellets here?

Response: The number 42% in the figure caption was a remnant of an earlier version of this graph and we apologize for this mistake. The respective number was removed from the caption.

As suggested, we have included the potential cycle for type 2 salp FP in Figure 2.

Comment 1.25: Suppl. table 1 – would be great with station depths included. Not shown (as far as I see)

Response: The respective station depths are now shown in supplementary Table 1. We have also adjusted the drifting trap labels to be consistent with the remaining manuscript.

Drifting trap	Station	Date	Time	Latitude	Longitude	Station depth	Comment
DF 01	118_01	25-04-2018	18:53	60°59.116'S	054°57.422'W	591.4	Deployment
DF 01	118_08	26-04-2018	10:54	60°55.622'S	055°09.133'W	345.8	Recovery

DF 02	119_01	26-04-2018	15:25	60°59.765°S	054°37.844'W	571.7	Deployment
DF 02	119_15	27-04-2018	12:02	60°55.899'S	054°43.898'W	667.6	Recovery
DF 03	120_01	27-04-2018	14:06	60°56.780'S	054°45.349'W	643.7	Deployment
DF 03	120_18	28-04-2018	10:55	60°57,450'S	054°52,048'W	640.7	Recovery
DF 04	121_01	28-04-2018	13:02	60°58,114'S	054°53,030'W	632.8	Deployment
DF 04	121_13	29-04-2018	11:00	60°58.948'S	054°55.870'W	607.5	Recovery
DF 05	122_01	29-04-2018	13:03	60°59.285'S	054°57.118'W	599.5	Deployment
DF 05	122_12	30-04-2018	11:00	60°57.406'S	054°58.179'W	980.2	Recovery

Comment 1.26: Suppl. Table 2 – is the mean size given the length, the width, or the ESD? Is the salp exp for Type 1 FP? Why isn't there data on the carbon content of FP per volume for salp FPs?

Response: The mean size (mm) given in Suppl. Table 2 (now Suppl. Table 3) refers to the mean length of krill and salps, respectively. We thank the reviewer for noticing that this information was missing. We have changed the table caption to include the missing information, and have adjusted the header of the respective column of the table: "...The mean size of krill and salps for each experiment is given in mm. For salps, the oral-atrial length was measured. Krill length was measured as total length following the AT method by Mauchline, 1980."

The salp pellet production experiments were conducted using surface water, thus the pellets produced were Type 1 pellets. Type2 salp FP (together with type 1 FP) were only collected by the drifting traps or were produced by salps collected at the surface at night. We have added a sentence to the table caption to clarify the type of salp pellets shown in this table: "...Salp FP in the experiments were produced by freshly caught salps and were conducted using surface in situ water. Thus, the pellets referred to in this table are Type 1 FP."

We have now included the mean and SD for the carbon content of the available salp FP in the table.

Comment 1.27: Suppl. Fig s7: I struggle to understand what DF they refer to, as there were 5 deployments and are 6 plots here? Is it the right or the middle lower panel that describes the high krill densities observed at the "last DF station"? The middle has a scale to 10 000 ind m⁻², while the right panel show most points (at a scale to 4 ind m⁻²...).

Response: Supplementary Figure 7 shows the time period during which the five drifting traps were deployed, which spanned six consecutive days. We agree with the reviewer that this illustration is confusing with respect to the single drifting traps. We have changed the figure to show the five drifting traps, one per panel, also in accordance with the comment by the reviewer to increase the consistency of the station names. We think that the updated version of the figure also better represents the high abundance of krill during the last drifting trap deployment (DF 05).

Supplementary Figure 7:

Reviewer #2 (Remarks to the Author)

Nature Communications manuscript NCOMMS-21-10982

Review of Pauli et al: " Shifts in krill and salps alter the efficiency of carbon export in the Southern Ocean"

In this study the authors combine a multitude of measurements, such as of krill and salp abundance together with sediment traps and with experiments of fecal pellet production of both krill and salp. Among other things they measure the carbon content and microbial degradation of fecal pellets and present the many different results this experimental setup produces. A key result of this work is that while salps produce more fecal pellets that sink faster than krill fecal pellets, these fecal pellets seem to be retained longer in the upper 300m and hence contribute less to the carbon export to 300m. The authors speculate that declining krill stocks may make the Southern Ocean a less efficient CO₂ sink in the future ocean.

This study is quite extensive; utilizing and combining different approaches and extensive analysis, all of which are satisfyingly documented in the manuscript. The approach chosen is valid and robust, and a natural step forward after the 2017 Iverson et al. paper in DSRII found that although salps produced fast sinking fecal pellets these pellets were retained surprisingly long in the upper water column. They suggested break-up and loosening of the pellets by zooplankton as the main cause, which is also one major conclusion of the current paper.

The paper is well written, particularly the Introduction, however with progression of the paper it gets harder and harder to follow it, possibly because of the many results and conclusions that need to be covered. I recommend the following revisions of the paper:

We thank the reviewer the helpful comments. We think that the suggestions have greatly improved our manuscript. Please see our responses to the specific comments below for details.

Comment 2.1: I suggest to highlight better the novelty of this study. The authors mention in line 81 that “Studies directly comparing the contribution of salp and krill FP are lacking.” But because the results of this study support/are similar to previous publications (e.g. Iverson et al., 2017 and others) it is easy to overlook this study is more complex etc. This should be stated ore clearly in the text.

Response: We have edited the introduction to better highlight the novel methodological approach of our study and to better distinguish this study from previous ones:

“Previous studies have used modelling approaches, sediment traps or moorings, and/or measurements of FP production and FP sinking velocities to estimate the FP carbon flux^{14,31,45}. In addition, the use of viscous gels in sediment traps (hereafter gel traps), which preserve the shape, size and structure of sinking particles, in combination with traditional sediment traps and in-situ particle camera systems, have provided high resolution vertical profiles of particles sizes and abundance^{6,35}. The use of gel traps also preserved salp FP³¹, which are fragile and easily break apart in conventional sediment traps, preventing direct flux estimates for salp FP. However, a combination of these in-situ approaches with measurements of biomass, FP production, sinking velocities, and microbial degradation is needed to provide better estimates of the extent of FP carbon flux. This study provides the first direct comparison of the contribution of krill and salp FP to the total organic carbon flux along Elephant Island at the northern tip of the Antarctic Peninsula.” (Lines 93-104).

Comment 2.2: My main concern with this study is that the abundance of salp in this study is measured very differently than that of krill: net hauls vs. acoustic measurements. While I understand why this is done, and it is well explained in lines 362 onwards, it is a big problem when comparing the results for krill and salp as it introduces biases, as the authors themselves mention. In a study like this

I would expect a more differentiated discussion of this bias. How big of a bias is introduced here, what does that mean for your results? Could that lead to higher/lower estimates of salp concentrations and/or fecal pellet production? It will give you an uncertainty margin that should be defined. This point to me is one of the weak points of this study and should be much better discussed.

Response: We understand the reviewers concern. For the highly mobile krill, net avoidance and escape are an issue in net sampling (e.g. Wiebe et al. 2003, Siegel 2016). In contrast, salps do not show net avoidance behavior and net sampling therefore provides a realistic representation of salp biomass. During our study, we sampled salps over multiple consecutive days and nights and thus have a good representation of the average salp abundances. We have now added a table to the supplementary material showing all net hauls conducted at the study site (Supplementary Table 2). For krill, acoustic surveys to determine krill density currently are the most up to date method (e.g. Watkins & Bierley 2002, Fielding et al. 2012), while a potential underestimation of the krill density may result from the so called “surface exclusion zone” due to the draft of the research vessel. We have added a paragraph discussing the potential bias resulting from the different methods used to the discussion section.

“The different methods used to assess the abundance of krill and salps may have resulted in an underestimation of the salp abundances that could potentially result in a lower estimated salp FP production. Net hauls using a Multiple Rectangular Midwater Trawl (Multi-RMT) between the surface and 330 m depth confirmed that salps were performing diel vertical migration to the surface at night and to depth during day (Supplementary Figure 6) and therefore confirms that we were able to catch the salps that migrated to the surface at night by sampling the upper 170 m.” (Lines 402-408)

Comment 2.3: Further, the paper mentions repeatedly that krill cause extreme events and at other times are absent or abundance is low. And salp adapt quicker to environmental changes (food availability) and are a more continuous factor in the study area if I understand correctly. So what does that mean for the results presented here? This somehow ties into my comment above. Firstly, one would expect that salps would start to produce fecal pellets faster if food increases possibly changing the numbers calculated. Is it possible to assess this? Secondly, you are most likely not catching any very large aggregations of salp with the net hauls such as the very dense krill swarms which are detected by the acoustics. I’d like to see a better discussion of these points.

Response: The gut passage time of 4-8 hours of salps was measured on-board during this study and confirmed previous observations (e.g. by Pakhomov 2004, DSRP II). We did not observe any differences in the fecal pellet production rates of salps between experiments conducted during day and night. Thus, we think that an increased food availability would not lead to a faster fecal pellet production. We agree with the reviewer that increased plankton availability could potentially lead to a higher carbon content of the produced FP. Yet, it was hypothesized that the feeding net of salps gets clogged when salps are exposed to high chlorophyll concentrations ($> 1 \text{ mg m}^{-3}$, cf. Perissinotto & Pakhomov 1997 Mar. Biol).

Salp swarms are 1-2 orders of magnitude smaller than krill swarms (10 vs. $>1,000 \text{ individuals m}^{-3}$). Therefore, with a mouth opening of the applied IKMT net of about 7 m^2 (HYDROBIOS), we think that we were able to capture also larger salp aggregations. In addition, salps do not show net avoidance behaviour as it has been observed for krill. Based on our sampling effort for salps, we think that our data show a realistic representation of the salp density during our study period. Please also see our response to the comment above.

Comment 2.4: The title of the paper is not ideal. In this paper I could not find any discussion on what the shift of krill and salp expected in a future ocean actually means in numbers and in relation to the results presented. I can only find a general, final paragraph hypothesizing possible changes and

unfortunately no calculation on how the efficiency will change in the future. Therefore, the title is misleading and should be changed.

Response: We agree with the reviewers' comments and have changed the title accordingly. The geographic region mentioned in the title was furthermore changed from Southern Ocean to Antarctic Peninsula in response to a comment by reviewer #1 to better highlight the regional aspect of this study: "Krill and salp faecal pellets contribute equally to the carbon flux at the Antarctic Peninsula"

Comment 2.5: I find there is some sloppy citing. While randomly checking some references, I found that in line 91 the citations 11 and 14 do not fit here. None of these papers claim krill FP are more efficient in carbon transport. You most likely mixed them up with other papers you reference. Please carefully check all your references.

Response: We apologize for any inconsistencies in the citations. All citations were carefully checked. The respective sentence mentioned by the reviewer was edited as follows:

"Thus, intact krill FP are more frequently found in sediment traps³⁴, suggesting that they are transferred to deeper water layers at high efficiency¹⁴ and thus are more efficient in carbon export than salp FP." (Lines 88-90)

Reviewer #3 (Remarks to the Author)

The manuscript presents the main findings of a study that assessed the relative contribution of krill and salp faecal pellets (FP) to vertical carbon flux during the cruise PS112 of the RV Polarstern off Elephant Island 383 (61°08'07.7"S, 55°12'14.9"W) conducted at the northern tip of the Antarctic Peninsula in April 2018. Results of the study indicate that on average, salps produced four-fold more FP carbon compared to krill, but the FP from both species contributed equally to the carbon flux to 300 m. The faecal pellets of the krill were efficiently exported to 300 m, while the fragile salp FP had strong retention due to fragmentation in the mixed layer. Decreased carbon fluxes associated with declining krill stocks and retention of salp FP indicate that the Southern Ocean could become a less efficient carbon sink for anthropogenic CO₂ in the future.

Overall, I found the manuscript to be well written and easy to read. Below I have listed a number of comments/suggestions that the authors should consider before the manuscript can be considered for publication.

We thank the reviewer for the helpful comments. Please see our responses to the specific comments below for more details.

Comment 3.1: I would encourage the authors to revise their abstract to include some actual data. Also, it would be useful if the authors could include information on where and when the study was conducted.

Response: The abstract was revised accordingly to include more concrete data, as well as information on the study region:

“Krill and salps are important for carbon flux in the Southern Ocean, but the extent of their contribution and the consequences of shifts in dominance from krill to salps remain unclear. We present a direct comparison of the contribution of krill and salp faecal pellets (FP) to vertical carbon flux at the Antarctic Peninsula using a combination of sediment traps, FP production, carbon content, microbial degradation, and krill and salp abundances. Salps produced 4-fold more FP carbon than krill, but the FP from both species contributed equally to the carbon flux at 300 m, accounting for 75% of total carbon. Krill FP were exported to 72% to 300 m, while 80% of salp FP were retained in the mixed layer due to fragmentation. Thus, declining krill abundances could lead to decreased carbon flux, indicating that the Antarctic Peninsula could become a less efficient carbon sink for anthropogenic CO₂ in future.” (Lines 25-34)

Comment 3.2 Line 48: Grazing by zooplankton may increase vertical carbon flux through diel vertical migrations and their grazing activities which produce faecal pellets which sink to depth.

Response: We have edited the respective paragraph according to the reviewers' comment:

“The efficiency of the BCP is driven by sinking organic aggregates, while fragmentation and grazing by zooplankton and microbial remineralisation processes decrease the efficiency of the BCP⁶. One major constituent of sinking aggregates are zooplankton faecal pellets (FP), which sink at high velocities and can make up the vast majority of sinking particles locally and therefore play a crucial role in carbon export^{7,8}. In addition, vertical migrations of zooplankton and the subsequent production of FP below the mixed layer contributes to the export of carbon⁹.” (Lines 44-50)

Comment 3.3. Line 56: The sentence is grammatically incorrect and needs to be revised.

Response: The sentence was rephrased and split into two parts:

“The contribution of krill FP to the carbon flux varies and ranges up to 281 mg C m⁻² d⁻¹¹³. At the Western Antarctic Peninsula (WAP) and the marginal ice zone krill FP were shown to account for 17–72% of the total carbon flux^{14,15}.” (Lines 54-56)

Comment 3.4. Line 67: Please check if the actual increase in seawater temperature increase 60C.

Response: We have checked the cited reference (Ducklow et al. 2007 *Phil. Trans. R. Soc. B*), where it is stated that “The Antarctic Peninsula is one among the most rapidly warming regions on Earth, having experienced a 2°C increase in the annual mean temperature and a 6°C rise in the mean winter temperature since 1950.”

The respective sentence in the introduction section was accordingly phrased as “Over the past decades, the WAP region has experienced dramatic climatic changes, including a 6 °C increase in mean winter air temperature since 1950, causing a 10% decline in sea ice extent^{20,21}...” (Lines 64-66)

Comment 3.5: Given the importance of phytoplankton size in mediating the partitioning of carbon between krill and salps, do the authors have any data on the size structure of the phytoplankton community during the period of study?

Response: Unfortunately, we only have data on the taxonomic composition of the plankton community during the study period available, which have been submitted as part of a separate manuscript (Pauli et al. under review, *Commun. Biol.*), which is referred to in Line 122 “Dinoflagellates and diatoms dominated the ambient plankton community³⁶...”

The taxonomic composition of the plankton community was studied using molecular methods (18S metabarcoding of variable region V4), which is not suitable to reliably assign taxa on species or genus level and the analysis was consequently conducted on the taxonomic level of Class. Therefore, a size structure cannot be reliably deduced from these data. We think that not only the size structure, but also the taxonomic community composition is a key factor in the partitioning of carbon.

Comment 3.6: It is likely that the net tows conducted only to a depth of 170m during the daytime sampling would have substantially underestimated the krill abundance and biomass. Indeed, it is worth noting that the discrepancy in krill abundances and biomass during the daytime and night-time samples is highlighted in Table 2. How might this underestimation have impacted the overall results of the investigation?

Response: For clarification, we would like to note that the net tows to a depth of 170 m were only conducted for salps, while for krill a hydroacoustic survey was used and calibrated against net tows. As the reviewer pointed out, it is correct that during daytime the abundance of salps at depths between 200 and 330 m was higher than at the surface due to the vertical migration patterns. This was also confirmed from net hauls using a Multiple Rectangular Midwater Trawl (8+1) and as shown in Supplementary Figure 6. The differences in the krill and salp abundances between day and night as shown in Table 2 are mentioned in the results section in lines 130 and 136, respectively:

“(Salp) Abundances were higher at night (278.98 ± 87.7 Ind. m^{-2}) than during daytime (93.96 ± 149.3 Ind. m^{-2} ; Table 3).”

“Krill abundance integrated over the top 200 m was 189.66 ± 182.4 Ind. m^{-2} at night and 667.9 ± 1396.2 Ind. m^{-2} during daytime (Table 3).”

In addition, we have added a paragraph discussing the potential bias introduced due to the different integration depths for krill and salps:

“Another potential bias in the calculation of the export efficiency of salp FP may result from the different integration depths used for krill and salp FP due to the different methods applied to measure abundance (net hauls vs. hydroacoustics). Estimating the FP production by salps integrated for the upper 200 m results in 70.14 mg C $m^{-2} d^{-1}$ (vs. 59.62 mg C $m^{-2} d^{-1}$ for the upper 170 m), which is 3% less than the export efficiency calculated based on the integrated FP production for the upper

170 m ($20 \pm 8.1\%$). Therefore, we assume that the resulting bias does not have a significant effect on the conclusions drawn.” (Lines 317-323)

Comment 3.7: I would encourage the authors to include the range of values for the different measurements in the results section rather than a single value.

Response: Throughout the results section, we have adjusted the values to show the range of the data, e.g. in line 118: “The average integrated standing stock of chlorophyll ranged from 64.8 to 115.8 mg m^{-2} and from 102.7 to 184.9 mg m^{-2} for the 0–100 m and 0–200 m depth layers...” . Alternatively, the mean value with standard deviation is shown, e.g. in line 120: “Similarly, the standing stock of particulate organic carbon (POC) was concentrated in the top 200 m with an integrated average of $12.45 \pm 3.6 \text{ g m}^{-2}$.”

Comment 3.8: It is evident from Table 1 that only a single estimate of primary production was conducted during the survey. Given the inherent variability in daily primary production rates in the Southern Ocean, are the authors convinced that their estimates of the percentage of primary production exported by the salps and krill to depth are realistic?

Response: We agree with the reviewers’ remark. We have primary production data available for two additional stations at Elephant Island, as well as for several other stations around the Antarctic Peninsula. However, these data were not collected during the same time period as the drifting trap deployments. The mean primary production during the entire cruise period in April 2018 was $52 \text{ mg C m}^{-2} \text{ d}^{-1}$ with a maximum value of $136 \text{ mg C m}^{-2} \text{ d}^{-1}$, which is in accordance with previous measurements during autumn at the WAP (cf. Vernet et al. 2012, MEPS: $102.3 \text{ mg C m}^{-2} \text{ d}^{-1}$). This would result in 43 to 114% of PP being export by salp FP, and 36 to 95% for krill. Thus, our estimate of the exported PP for this particular region and season rather represents an underestimation. We have added a sentence on the regional variability in primary production during our study to the results section: “Primary production at 20 m in the study area at Elephant Island was $20.62 \text{ mg C m}^{-3} \text{ d}^{-1}$. In comparison, the mean primary production across different areas around the northern Antarctic Peninsula in the same month (04/2018) was $10.4 \pm 9.5 \text{ mg C m}^{-3} \text{ d}^{-1}$.” (Lines 123-126)

Comment 3.9. Line 425: “Faecal pellet production rates of krill and salps were measured from incubation experiments using freshly caught animals.” Please clarify if these samples were collected during the daytime/night time since this may have influenced the amount of pigment in the guts of the krill.

Response: We agree with the reviewer that the amount and composition of phytoplankton consumed during day and night respectively might influence the carbon content of the produced fecal pellets. However, we did not observe differences in the fecal pellet production of salps between day and night. We have now added information whether krill and salps for the incubation experiments were collected at day or night to Supplementary Table 3 (formerly Supplementary Table 2). In addition, we have adjusted the respective sentence in the Material & Methods section: “Faecal pellet production rates of krill and salps were measured from incubation experiments using freshly caught animals during day and night (Supplementary Table 3).” (Line 483) However, as we did not use gut fluorescence methods in this study, therefore we think that the amount of pigments is of minor importance in this context.

Comment 3.10. Line 507: “The pellet volume was calculated assuming an ellipsoid shape for salps, and a cylindrical shape for krill pellets.” Please clarify or include a reference.

Response: The sentence was edited to clarify the method and references were added: “Based on the different geometric shapes of krill and salp FP, the FP volume was calculated assuming an ellipsoid

shape for salp, and a cylindrical shape for krill pellets, as used in previous studies^{15,31}.” (Lines565-567)

Comment 3.11: Figure 4 of the manuscript is redundant and can be omitted from the manuscript.

Response: We think that Figure 4 provides a good visualization of the different shapes of salp FP and their fragile structure that is not represented in any of the other figures of the manuscript. Figure 4 is also complementary to Supplementary Figure 9 showing the changes in shape of krill FP with depth. We further find that it illustrates the preservation of salp FP in gel traps well. It was highlighted by reviewer #1 that figures including pictures are appreciated. The journal allows up to 10 displayed items; therefore, we would like to keep Figure 4 in the manuscript.

REVIEWER COMMENTS

Reviewer #2 (Remarks to the Author):

Nature Communications manuscript NCOMMS-21-10982

Review of the revised manuscript by Pauli et al: " Krill and salp faecal pellets contribute equally to the carbon flux at the Antarctic Peninsula"

The manuscript has been extensively revised to include the comments of three reviewers. That means that a lot of points have been clarified and the manuscript is much improved now. Most of my comments, including my request to revise the paper title, have been addressed satisfactorily.

Concerning my major concern about this paper: that the abundance of salp in this study is measured very differently than that of krill: net hauls vs. acoustic measurements (comment 2.2), I am slightly disappointed by the authors reply. As stated before, I understand why this is done, and I am not criticizing the approach as such. But when comparing the results for krill and salp this introduces biases, as the authors themselves mention in the ms. I had requested a discussion on how much of a bias is introduced that was not answered quite as detailed as I had hoped.

Therefore, would like to ask again if it is not possible to define how big this introduced bias is? The authors included an additional paragraph in the discussion in response to my comment 2.2, which states that this "may have resulted in an underestimation of the salp abundances that could potentially result in a lower estimated salp FP production." That is good to know and along the lines of what I expected but I am wondering if it is not possible to roughly estimate (back-of-the-envelope calculation) this bias to tell the reader that the salp abundance is underestimated by ~1%, 10% or 50%, 80%? And state what that means for fecal pellet production estimates? It is important for the reader to know this bias to be able to appreciate the conclusions of this study better and will strengthen the impact of this manuscript.

I conclude that this paper has gained a lot through the review process and in my eyes is ready for publication once my last point has been clarified. I congratulate the authors on this very extensive study and important contribution to research on carbon export at the WAP.

Reviewer #3 (Remarks to the Author):

I have been through the revised manuscript and the response to the reviewers document and am satisfied that the authors have largely addressed the main concerns of the reviewers. The authors are to be congratulated on a job well done. In my opinion, the manuscript can now be accepted for publication.

Nature Communications manuscript NCOMMS-21-10982A – Response to reviewers

REVIEWER COMMENTS

Reviewer #2 (Remarks to the Author):

Nature Communications manuscript NCOMMS-21-10982

Review of the revised manuscript by Pauli et al: "Krill and salp faecal pellets contribute equally to the carbon flux at the Antarctic Peninsula"

The manuscript has been extensively revised to include the comments of three reviewers. That means that a lot of points have been clarified and the manuscript is much improved now. Most of my comments, including my request to revise the paper title, have been addressed satisfactorily.

Concerning my major concern about this paper: that the abundance of salp in this study is measured very differently than that of krill: net hauls vs. acoustic measurements (comment 2.2), I am slightly disappointed by the authors' reply. As stated before, I understand why this is done, and I am not criticizing the approach as such. But when comparing the results for krill and salp this introduces biases, as the authors themselves mention in the ms. I had requested a discussion on how much of a bias is introduced that was not answered quite as detailed as I had hoped.

Therefore, I would like to ask again if it is not possible to define how big this introduced bias is? The authors included an additional paragraph in the discussion in response to my comment 2.2, which states that this "may have resulted in an underestimation of the salp abundances that could potentially result in a lower estimated salp FP production." That is good to know and along the lines of what I expected but I am wondering if it is not possible to roughly estimate (back-of-the-envelope calculation) this bias to tell the reader that the salp abundance is underestimated by ~1%, 10% or 50%, 80%? And state what that means for fecal pellet production estimates? It is important for the reader to know this bias to be able to appreciate the conclusions of this study better and will strengthen the impact of this manuscript.

Response: We are sorry that the reviewer was not satisfied with our previous attempt to address this comment. We had not provided concrete percentages for possible underestimates of salp abundance, as we did not see a solid basis to quantify exactly by what percentage salp abundances might have been underestimated. However, according to the reviewers' recommendation, we have attempted to give the reader a more concrete idea on how the export efficiency of salp pellets would be affected, if we would assume different salp frequency scenarios. For this, we have calculated how a potential underestimation of the salp abundance by 10% and 50% would have affected the export efficiency of salp pellets compared to our regular scenario (0%). We have added a table to the supplementary material and have extended the respective paragraph in the discussion section (Lines 404-418):

"The different methods used to assess the abundance of krill and salps may have resulted in an underestimation of the salp abundances that could potentially result in a lower estimated salp FP production. However, we expect this underestimation to be minor, as salps do not show escape behavior from plankton nets. Assuming we underestimated the salp abundance and subsequent FP production by 10%, the resulting export efficiency of salp FP to 300 m would be 18.2% instead of 19.9% (Supplementary Table 5) and would therefore not influence the main conclusions drawn here. If the underestimation of salp abundance and FP production was 50%, the resulting export efficiency would be considerably less (13.3%). Thus, even a 50% underestimation, which is very unlikely, would still mean that salp pellets are efficiently turned over in the upper water column. Moreover, in this study, net hauls using a Multiple Rectangular Midwater Trawl (Multi-RMT) between the surface and 330 m depth

confirmed that salps were performing diel vertical migration to the surface at night and to depth during day (Supplementary Figure 6) and therefore confirm that we were able to catch the salps that migrated to the surface at night by sampling the upper 170 m.”

Supplementary Table 5. Potential FP flux of salps and resulting export efficiency to 300 m under the regular scenario, and under the assumption that salp abundance and FP production were underestimated by 10% and 50%, respectively.

		Regular	+ 10%	+ 50%	Unit
Potential FP Flux > 170 m	Day	19.84	21.83	29.76	mg C m ⁻² d ⁻¹
	Night	39.78	43.75	59.66	mg C m ⁻² d ⁻¹
	24 h	59.62	65.58	89.43	mg C m ⁻² d ⁻¹
Export efficiency to 300m		19.99 ± 8.10	18.17 ± 7.36	13.33 ± 5.40	%

I conclude that this paper has gained a lot through the review process and in my eyes is ready for publication once my last point has been clarified. I congratulate the authors on this very extensive study and important contribution to research on carbon export at the WAP.

Response: We are very thankful to the reviewer for the helpful comments, which significantly improved our manuscript.

Reviewer #3 (Remarks to the Author):

I have been through the revised manuscript and the response to the reviewers document and am satisfied that the authors have largely addressed the main concerns of the reviewers. The authors are to be congratulated on a job well done. In my opinion, the manuscript can now be accepted for publication.

Response: We would like to thank the reviewer again for the valuable comments that have greatly improved our manuscript.